# Mechanisms of PDZ domain scaffold assembly illuminated by use of supported cell membrane sheets

Simon Erlendsson[1,2†], Thor Seneca Thorsen[1†], Georges Vauquelin[3], Ina Ammendrup-Johnsen[1], Volker Wirth[4], Karen L Martinez[4], Kaare Teilum[2], Ulrik Gether[1], Kenneth Lindegaard Madsen[1*]

[1]Molecular Neuropharmacology and Genetics Laboratory, Department of Neuroscience, Faculty of Health and Medical Sciences, The Panum Institute, University of Copenhagen, Copenhagen, Denmark; [2]Structural Biology and NMR Laboratory, Department of Neuroscience, University of Copenhagen, Copenhagen, Denmark; [3]Molecular and Biochemical Pharmacology, Department of Biotechnology, Free University Brussels (VUB), Brussels, Belgium; [4]Bionanotechnology and Nanomedicine Laboratory, Department of Chemistry, Nano-science Center, University of Copenhagen, Copenhagen, Denmark

*For correspondence:
lnp353@ku.dk

[†]These authors contributed equally to this work

Competing interests: The authors declare that no competing interests exist.

**Abstract** PDZ domain scaffold proteins are molecular modules orchestrating cellular signalling in space and time. Here, we investigate assembly of PDZ scaffolds using supported cell membrane sheets, a unique experimental setup enabling direct access to the intracellular face of the cell membrane. Our data demonstrate how multivalent protein-protein and protein-lipid interactions provide critical avidity for the strong binding between the PDZ domain scaffold proteins, PICK1 and PSD-95, and their cognate transmembrane binding partners. The kinetics of the binding were remarkably slow and binding strength two-three orders of magnitude higher than the intrinsic affinity for the isolated PDZ interaction. Interestingly, discrete changes in the intrinsic PICK1 PDZ affinity did not affect overall binding strength but instead revealed dual scaffold modes for PICK1. Our data supported by simulations suggest that intrinsic PDZ domain affinities are finely tuned and encode specific cellular responses, enabling multiplexed cellular functions of PDZ scaffolds.
DOI: https://doi.org/10.7554/eLife.39180.001

## Introduction

It is of fundamental importance for cell function to organize signaling processes in space and time. Scaffold proteins play a key role in these efforts by operating as versatile nanoscale modules capable of bringing distinct molecular components in close proximity to shape specificity in cellular signaling networks and regulate output (*Good et al., 2011*; *Zeke et al., 2009*). A broad variety of different protein-protein and protein-lipid interacting domains are found in scaffold proteins, enabling them to bind and direct localization and function of their diverse interaction partners, such as receptors, transporters, ion channels and kinases (*Hung and Sheng, 2002*; *Zhu et al., 2016*).

Our current understanding of the dynamics and kinetics underlying scaffold interactions relies mostly on in vitro assays of single domains isolated form their native membrane environment (*Vincentelli et al., 2015*; *Stiffler et al., 2007*; *Long et al., 2003*; *Ivarsson, 2012*). Such approaches reduce the complexity and makes binding assays simpler to both perform and analyze. Nonetheless, the presence of several interaction domains in one protein, together with possible formation of higher order structures of both scaffold proteins and transmembrane interaction partners (*Long et al., 2003*; *Ivarsson, 2012*), strongly suggest that measurements in vitro cannot replicate

**eLife digest** Inside a cell, many different signals carry information that is essential for the cell to remain healthy and perform its role in the body. It is, therefore, very important that the signals are sent to the right places at the right times. Scaffold proteins play an essential role in organizing these signals by bringing specific proteins and other molecules into close contact at particular times and locations within the cell. Defects in scaffolding proteins can lead to cancer, psychiatric disorders and other diseases, so these proteins represent potential new targets for medicinal drugs.

Many scaffolding proteins assemble groups of proteins on the surface of the membrane that surrounds the cell. Previous studies have shown that scaffolding proteins are able to bind to several other proteins as well as the membrane itself at the same time. However, the precise way in which scaffolding proteins assemble such groups is not clear because it is technically challenging to study this process in living cells. To overcome this challenge, Erlendsson, Thorsen et al. used a new experimental setup known as supported cell membrane sheets – which provides direct access to the side of the cell membrane that usually faces into the cell – to study two scaffolding proteins known as PICK1 and PSD-95.

The experiments show that PICK1 and PSD-95 bind to their partner proteins up to 100 times more strongly than previously observed using other approaches. This is due to the scaffolding proteins binding more strongly to both their partners and the membrane.

Unexpectedly, the experiments show that the shape and physical characteristics of the partner protein have no effect on the increase in the strength of the binding. Further experiments suggest that altering the ability of the PDZ domain of PICK1 to bind to partner proteins changes the mode of action of the PICK1 protein so that it can activate different responses in the cell. Together these findings imply that the ability of scaffolding proteins to bind to their partner proteins is finely tuned to encode specific responses in cells in different situations – a hypothesis that Erlendsson, Thorsen et al. are planning to test in intact cells.

DOI: https://doi.org/10.7554/eLife.39180.002

the behavior of the native environment. Furthermore, scaffold proteins often function on, or in proximity, to lipid cellular membranes, and several scaffold protein domains directly interact with lipids (*Egea-Jimenez et al., 2016*; *Pérez et al., 2013*).

PDZ (PSD-95/Discs Large/ZO-1) domains constitute one of the most common interaction domains in scaffold proteins (*Feng and Zhang, 2009*; *Zhu et al., 2016*; *Ye and Zhang, 2013*) and are characterized by an elongated binding groove that interacts with the last three to five C-terminal residues of the target proteins (*Doyle et al., 1996*). Scaffold proteins often contain several PDZ domains permitting them to serve as adaptors to assemble protein complexes (*Doyle et al., 1996*; *Sheng and Sala, 2001*; *Feng and Zhang, 2009*; *Ye and Zhang, 2013*). PICK1 (Protein Interacting with C Kinase 1), for example, forms a dimeric structure containing two spatially separated, identical PDZ domains and a lipid binding N-BAR (Bin/amphiphysin/Rvs) domain (*Karlsen et al., 2015*), enabling simultaneous binding of two interaction partners and tethering of the complex to the membrane (*Xu and Xia, 2006*). To perform its differential functions, PICK1 is believed to operate in two distinct modes of scaffolding; one required for clustering transmembrane interaction partners, such as the AMPA-type glutamate receptors (AMPARs), the metabotropic glutamate receptor 7 (mGluR7), ephrin and the monoamine transporters (*Xia et al., 1999*; *Torres et al., 2001*; *Boudin et al., 2000*; *Torres et al., 1998*), and one required for recruiting protein kinase Cα (PKCα) to its transmembrane interaction partners to regulate their phosphorylation (*Baron et al., 2002*; *Dev et al., 2000*; *Perez et al., 2001*; *Staudinger et al., 1995*). The molecular mechanisms underlying these different scaffold modes remain nevertheless unknown.

In contrast to neuronal scaffold interactions, drug-receptor or antibody interactions have been studied rigorously in tissues and cells uncovering molecular mechanisms of co-operativity and avidity in many biological systems (*Varner et al., 2015*). To enable similar studies of scaffold interactions with membrane proteins embedded in their natural membrane environment, we take here advantage of a supported cell membrane sheets (SCMS) technique (*Perez et al., 2006a*; *Perez et al., 2006b*). SCMS are prepared by pressing a glass cover-slip on an adherent cell culture. When the

cover-slip is removed, the apical plasma membrane detaches from cells and a planar sheet exposing the inner surface of the membrane is exposed on the cover-slip. Binding of fluorescence labeled protein ligands to the membrane proteins exposed on the SCMS can be quantified by confocal laser scanning microscopy. Strikingly, our data reveal binding strengths for PICK1, as well as for the PDZ tandem domain from PSD-95 (*Long et al., 2003*; *Cho et al., 1992*), that are two to three orders of magnitude higher than the intrinsic affinities measured in vitro. The binding strength for PICK1 is strongly dependent on an intact PDZ domain binding groove and the membrane-binding N-BAR domain but, surprisingly, independent of the tertiary and quaternary structure of the transmembrane PDZ binding partners. To our further surprise, the binding strength for PICK1 is insensitive to discrete changes in the intrinsic affinities of the PDZ binding partners, which instead are reflected in changes of maximal binding. Mathematical modeling of homo-bivalent binding demonstrates that the observed behavior is consistent with a change in the binding mode from a scenario where one PICK1 dimer binds two membrane ligands to a scenario where one PICK1 dimer binds one membrane ligand and has one free PDZ domain available for other interactions. Altogether, by quantitatively illuminating binding kinetics of PDZ domain protein scaffolds on a cell membrane, our results reveal novel principles for how cellular scaffold proteins can operate to ensure specificity and selectivity in cellular signaling networks, and furthermore how these principles change our current understanding of cellular binding equilibria and mechanisms underlying the function of scaffold proteins in general.

## Results

### SCMS reveals nanomolar binding strength of scaffold interactions

To investigate the interaction between PICK1 and the GluA2 subunit of the AMPAR in a semi-native environment, N-terminally flag-tagged GluA2 (SF-GluA2) was transiently expressed in HEK293 cells and labeled with Alexa-488 conjugated anti-FLAG M1 antibody before preparation of supported cell membrane sheets (SCMS) (*Perez et al., 2006a*) (*Figure 1a–d*). We incubated SCMSs, exposing the inner membrane leaflet and the intracellular parts of SF-GluA2, with increasing concentrations of purified PICK1, containing an N-terminal SNAP-tag fluorescently labeled with SNAP-surface 549 (PICK1) (*Figure 1a,b*), and measured the fluorescence intensity of bound PICK1 by confocal microscopy. By normalizing the intensities of the fluorescent signal from PICK1 (red) to the intensity of fluorescently labeled SF-GluA2 (green) and plotting this ratio as a function of increasing PICK1 concentration, we obtained a saturable binding curve (*Figure 1f*). The apparent affinity ($K_d^*$) calculated from the binding curve was $67 \pm 6$ nM (mean $\pm$ s.e.m., n = 3), which, remarkably, is ~100 fold higher than the low micromolar intrinsic affinity ($K_d^{int}$) determined for binding of the GluA2 C-terminus to the PICK1 PDZ domain using an in-solution based assay (*Erlendsson et al., 2014*). To test if the observed binding was specific and dependent on the C-terminal PDZ binding motif (-ESVKI) in GluA2, we added an alanine residue to the GluA2 C-terminus (-ESVKI + A) (SF-GluA2 +A) to compromise PDZ ligand binding (*Madsen et al., 2005*; *Madsen et al., 2008*). The apparent affinity and the maximal binding ($B_{max}$) of PICK1 were significantly reduced compared to binding to SF-GluA2 for similar receptor expression levels (*Figure 1b,e,f*). This supports that PICK1 binding to SCMS from GluA2 expressing cells is specific and depends on the interaction with the GluA2 C-terminal PDZ motif. The results also establish the use of SCMS as a new, robust, quantitative method for investigating membrane proximal scaffold interactions.

### Membrane anchored protein tails are sufficient to enable strong binding of PICK1 on SCMS

To test if the increased binding strength measured in the SCMS assay, compared to the micromolar affinity measured by in-solution assays (*Erlendsson et al., 2014*), was dependent on the tetrameric arrangement of subunits within the AMPARs, we transferred the 24 C-terminal residues of GluA2 onto the C-terminus of the single transmembrane spanning α-subunit of the IL-2 receptor (TAC) fused to YFP (TAC-YFP-GluA2) (*Figure 1g*). In contrast to the AMPARs, TAC is not believed to form higher order structures (*Spangler et al., 2015*). Strikingly, we obtained a $K_d^*$ of $73 \pm 19$ nM (mean $\pm$s.e.m., n = 3) for binding of PICK1 to TAC-YFP-GluA2 (*Figure 1h,i*) suggesting that the high

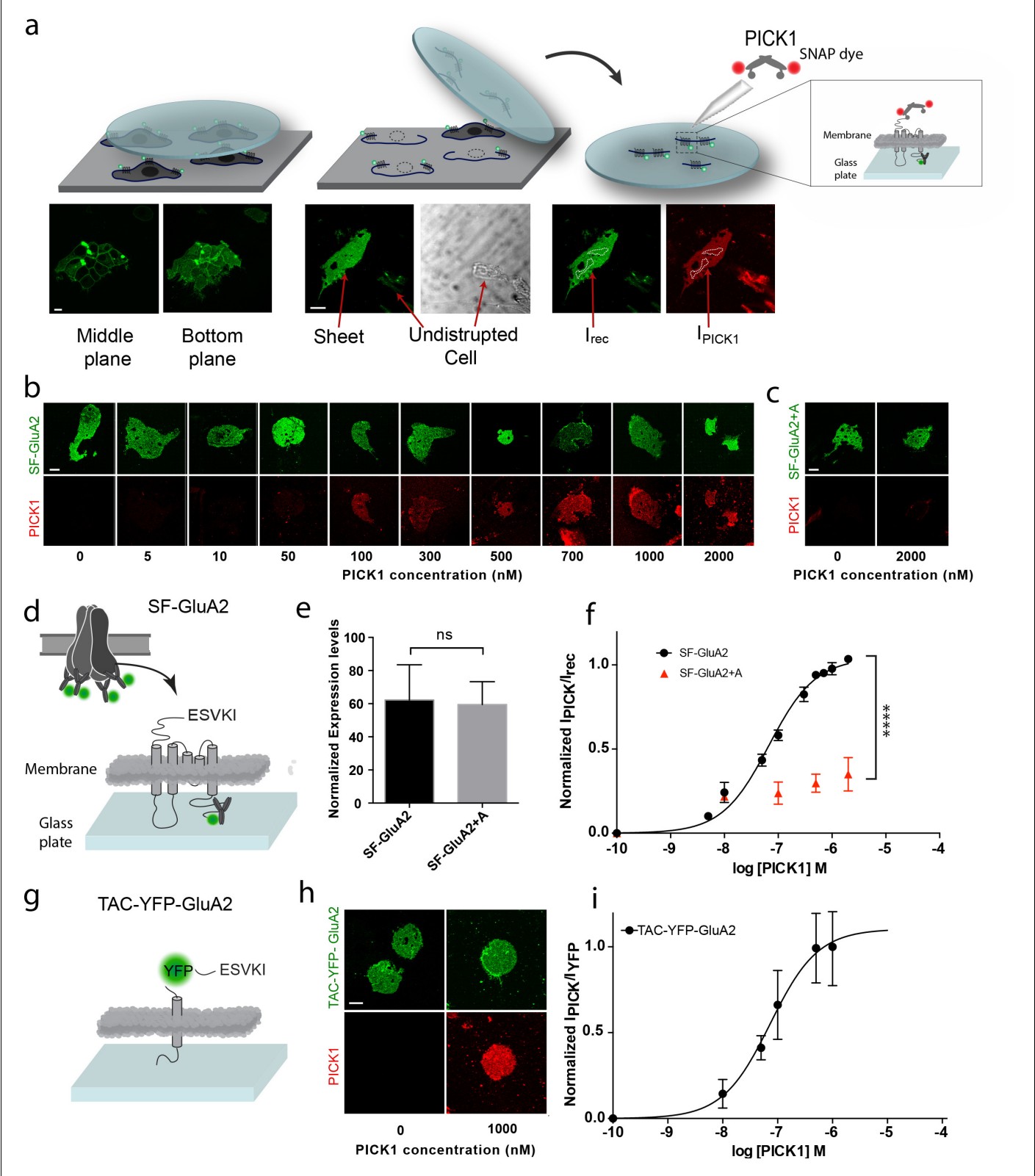

**Figure 1.** PICK1 PDZ binds with nanomolar binding strength to transmembrane interaction partners in semi-native membranes. (a) SCMS were prepared from HEK293 Grip tite cells transfected with a fluorescently labeled (green) membrane protein of interest (left) by pressing a pre-coated glass coverslip onto the cells (middle) and subsequently detaching the apical plasma membranes from the remains of the cells (right). Single bilayers were readily distinguished from undisrupted cells or organelles by their lack of three-dimensional structure. SCMS were incubated with buffer containing

*Figure 1 continued on next page*

*Figure 1 continued*

fluorescently labeled scaffolding protein (red). Binding was quantified by measuring the intensity of PICK1, $I_{PICK1}$, relative to the intensity of the receptor, $I_{rec}$, in a region of interest (dashed white lines). (b) Representative confocal images demonstrating concentration dependent binding of PICK1 (red) to SCMS expressing SF-GluA2 (green), or (c) endpoints for SF-GluA2 +A. (d) Schematic representation of SF-GluA2 constructs labeled with anti-FLAG M1-alexa 488 antibody. (e) Expression levels of SF-GluA2 and SF-GluA2 +A in SCMS were similar (p=0.66). (f) Quantification of concentration dependent binding of PICK1 on SCMS's expressing SF-GluA2 (black) ($K_d$*=67 ± 6 nM), or SF-GluA2 +A (red), (n = 3, ****p≤0.0001). (g) Schematic representation of TAC-YFP-GluA2 constructs (h) Representative end points of PICK1 concentrations series. (i) Quantification of concentration dependent binding of PICK1 to TAC-YFP-GluA2 ($K_d$*=73 ± 19 nM) (n = 3). Scale bars 10 μm.
DOI: https://doi.org/10.7554/eLife.39180.003

apparent affinity was achieved independently of the tetrameric complex as well as of the membrane embedded segments of the receptor.

To address how different PDZ-binding motifs affect the binding strength, we next measured the interaction of PICK1 with the C-terminus of the dopamine transporter DAT (TAC-YFP-DAT C24) (*Figure 2*). According to in-solution binding assays, this peptide has a 10-fold higher intrinsic affinity for PICK1 than the GluA2 peptide (*Erlendsson et al., 2014*). We observed specific binding also for this construct (*Figure 2—figure supplement 1*) but only a minor increase in $K_d$* (47 ± 5 nM, mean ±s.e.m., n = 7), revealing that the binding strength measured in the SCMS assay correlates poorly with the intrinsic affinity (*Figure 1a,c*).

To further confirm specificity and rule out effects of the SNAP-tag on PICK1, competition binding using a fixed concentration of labeled SNAP-PICK1 and an increasing concentration of unlabeled PICK1 was performed on SCMSs expressing TAC-YFP-DAT (*Figure 2b,d*). The binding strength determined from competition binding ($K_i$*=29 ± 5 nM (mean ±s.e.m., n = 5) was close to that from the direct binding saturation assay and increased 100-fold compared to the intrinsic affinity of the PICK1 PDZ interaction with the DAT C-terminus obtained in solution using a fluorescence polarization competition assay (2.1 ± 0.4 μM) (*Figure 2b,d*).

To complement the results for PICK1, we also probed the binding strength of fluorescently labeled PSD-95 PDZ 1–2 tandem domain on SCMS expressing the ß1-adrenergic receptor (*Hu et al., 2000*). Indeed, the measured binding strength ($EC_{50}$ = 3 ± 2 μM) was two orders of magnitude increased compared to the intrinsic affinities previously measured in solution for either of the two domains (430 ± 47 μM and 120 ± 20 μM for PDZ1 and PDZ2, respectively) (*Møller et al., 2013*) (*Figure 2—figure supplement 2a,b*). These findings further support that the binding strengths of PDZ domain scaffolding interactions are substantially higher than intrinsic PDZ affinities.

## Intrinsic affinities of PDZ ligands control maximal binding of PICK1

Despite similar high binding strength, comparison of the binding curves for PICK1 binding to TAC-YFP-GluA2 and TAC-YFP-DAT revealed a surprising two-fold difference in total maximal binding ($B_{max}$) (TAC-YFP-DAT $B_{max}$ = 100%; TAC-YFP-GluA2 $B_{max}$ = 44 ± 9%) (*Figure 2—figure supplement 3a,b*). Note that this difference unlikely is due to different expression levels as the maximum binding was normalized to the YFP signal for the two different constructs. To address whether the difference was a consequence of the different intrinsic affinities of the DAT C-terminus compared to the GluA2 C-terminus, we exploited that the intrinsic affinity of PICK1 for the DAT C-terminus depends on the C-terminal valine and that substitution of the aliphatic side-chain of the C-terminal valine decrease its intrinsic affinity for PICK1; Val (WT) (2.3 ± 0.1 μM)>Ile (9.5 ± 0.9 μM)>Ala (49 ± 3 μM) (*Madsen et al., 2005*). Because DAT plasma membrane targeting is compromised by alterations in the extreme C-terminus (*Bjerggaard et al., 2004*), we introduced the mutation series into a previously characterized fusion construct in which the DAT C-terminus is fused to β2 adrenergic receptor (flagβ2-DAT8) (*Madsen et al., 2012*) yielding three constructs: LKV, LKI and LKA, respectively (*Figure 3—figure supplement 1a*). The apparent affinity of PICK1 to LKV ($K_d$*=37 ± 5 nM, n = 8) (mean ±s.e.m.) (*Figure 3a,b*) was essentially the same as that seen for TAC-YFP-DAT (*Figure 2*). Moreover, despite a decrease in intrinsic affinity of up to >20 fold, we observed no differences in apparent affinity in the SCMS assay upon mutating the valine to isoleucine (LKI $K_d$*=39 ± 4 nM, n = 7) or alanine (LKA $K_d$*=59 ± 11 nM, n = 5) (*Figure 3a,b*). Instead, we observed an unexpected reduction in maximal binding ($B_{max}$ LKI: 56 ± 2%, LKA: 41 ± 4%; (means ±s.e.m.) relative to LKV

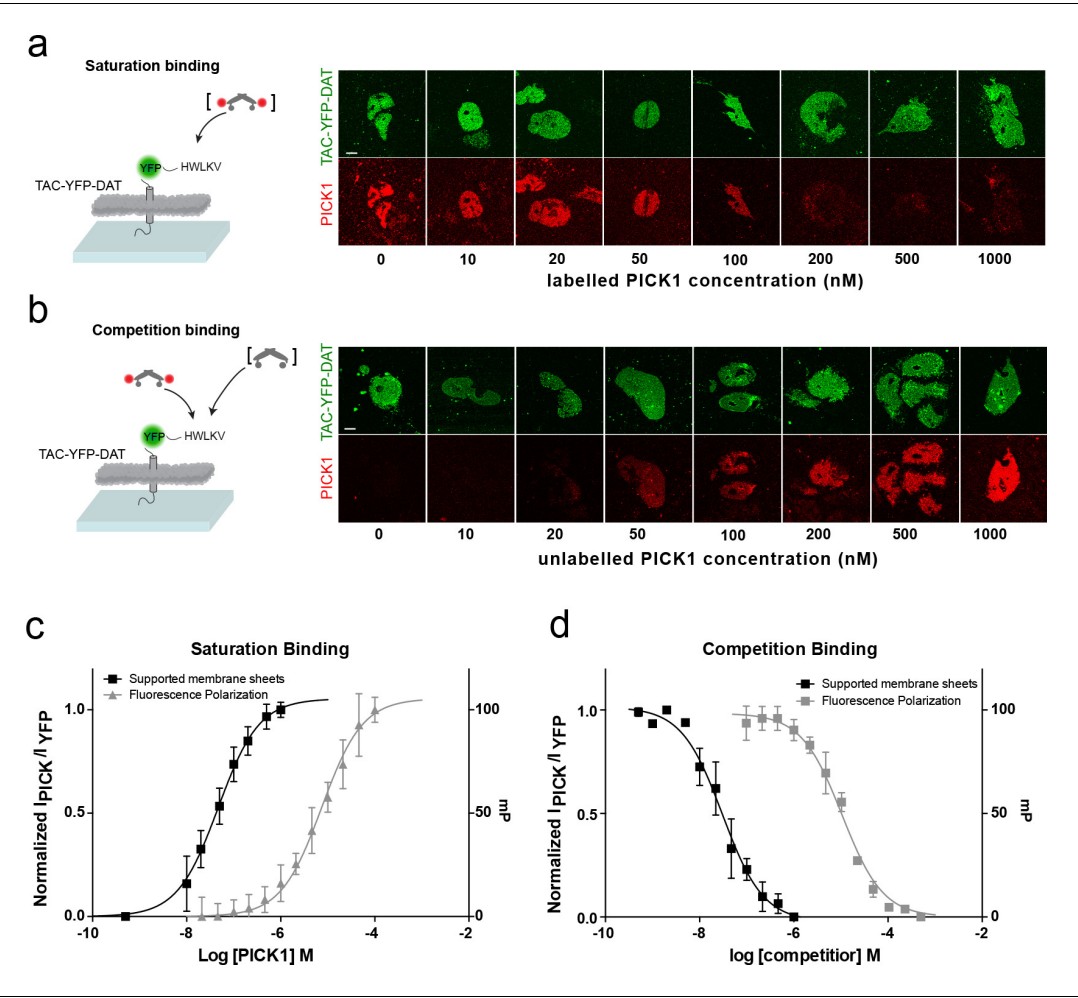

**Figure 2.** PICK1 binding on SCMS is specific and the binding strength is increased two orders of magnitude compared to in-solution affinities. Schematic and representative confocal images demonstrating saturation with labeled PICK1 (**a**) and competition between labeled PICK1 and unlabeled PICK1 (**b**) on SCMS expressing TAC-YFP-DAT. Brackets indicate the varied PICK1 pool. (**c**) Normalized binding as a function of unlabeled PICK1 concentration (black), left axis ($K_d^* = 47 \pm 5$ nM) (n = 7) compared to in-solution based FP-measurements (grey), right axis ($K_d = 9 \pm 2$ µM, n = 3, performed in triplicates). (**d**) Normalized binding as a function of unlabeled PICK1 concentration (black), left axis ($IC_{50} = 29 \pm 4$ nM) (n = 5) compared to in-solution based FP-measurements (grey), right axis ($K_{i,app} = 2 \pm 0.4$ µM, n = 3, performed in triplicates). Scale bars 10 µm.
DOI: https://doi.org/10.7554/eLife.39180.004

The following figure supplements are available for figure 2:

**Figure supplement 1.** The PICK1 – TAC-YFP-DAT interaction is saturable and depends primarily on the PDZ binding motif.
DOI: https://doi.org/10.7554/eLife.39180.005
**Figure supplement 2.** PSD-95 PDZ1-2 binding to SF-β1AR in supported cell membrane sheet.
DOI: https://doi.org/10.7554/eLife.39180.006
**Figure supplement 3.** The intrinsic PDZ affinity does not translate directly to avidity but determines maximal binding.
DOI: https://doi.org/10.7554/eLife.39180.007

(*Figure 3a,b* and *Table 1*) for similar receptor surface expression levels (*Figure 3—figure supplement 1b*).

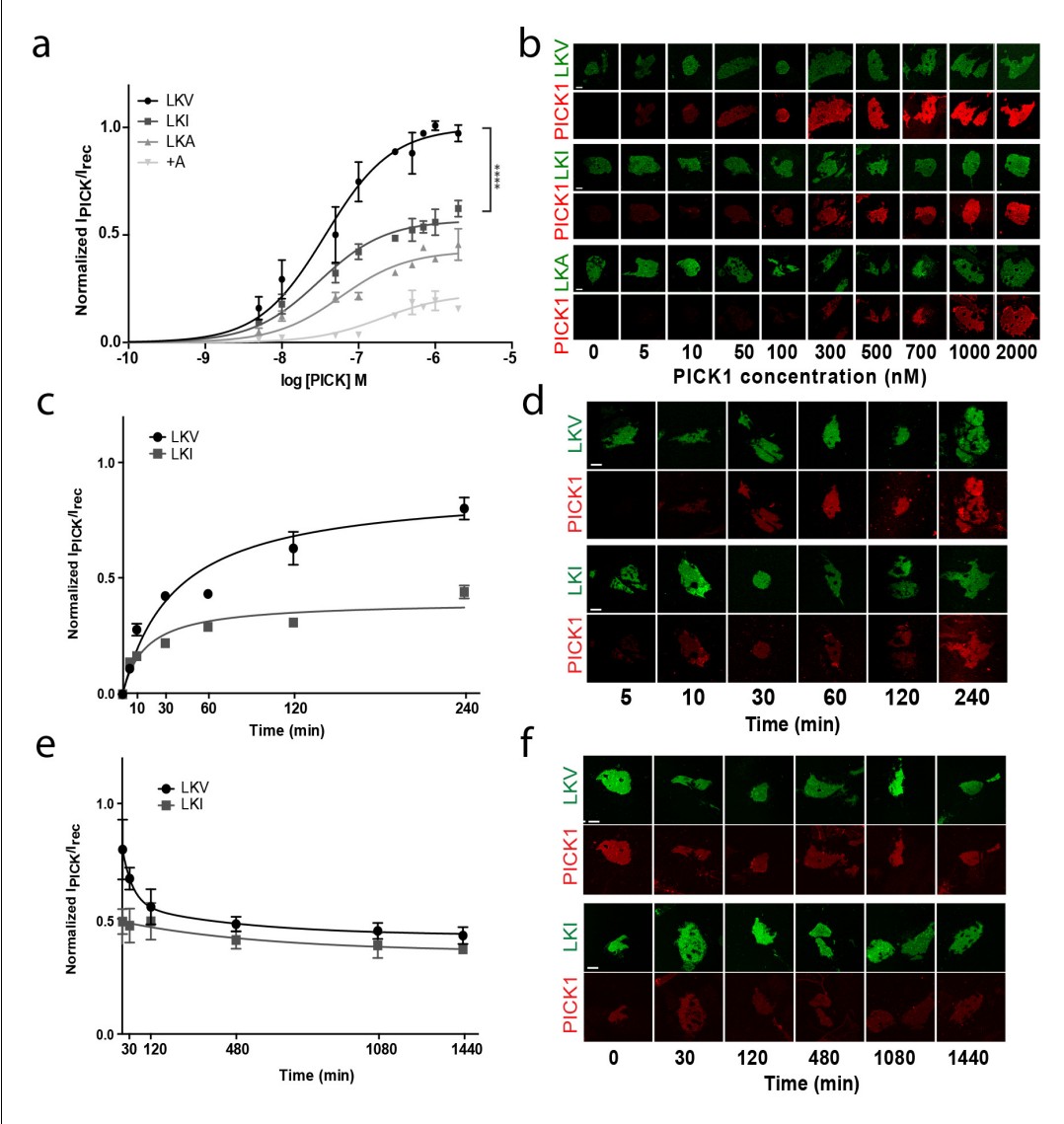

**Figure 3.** The intrinsic PDZ affinity does not translate directly to avidity but determines maximal binding. (**b**) Normalized binding derived from SCMS as a function of PICK1 concentration to LKV (black) ($K_d^*$=37 ± 5 nM, $B_{max}$ = 100%), LKI (dark grey) ($K_d^*$=29 ± 4 nM, $B_{max}$ = 56 ± 2%), LKA (light grey) ($K_d^*$=59 ± 11 nM, $B_{max}$ = 41 ± 4%) LKV +A (white) ($K_d^*$not fitted, $B_{max}$ = 21 ± 3). n = 8, 7, 5 and 3, respectively (**p<0.01; ****p≤0.0001). (**b**) Representative images demonstrating concentration dependent binding of PICK1 to SCMS expressing LKV, LKI and LKA constructs. (**c**) Representative PICK1 binding to SCMS expressing LKV or LKI as a function of incubation time. PICK1 concentration is 100 nM. Half maximum binding values are 24 ± 14 min for LKV, and 11 ± 4 min for LKI (means ±s.e.m, n = 3). (**d**) Representative images showing time dependent PICK1 binding to LKV and LKI. (**e**) Representative PICK1 dissociation curves from SCMS expressing LKV or LKI (points are means ±SD). LKV is fitted to a two-state dissociation with estimated fast and slow half-life of 21 ± 8 and 373 ± 51 min., respectively. LKI is fitted to a one-state dissociation with a half-life of 431 ± 16 min. (means ± S.E, n = 3). (**f**) Representative images showing time dependent PICK1 dissociation from LKV and LKI. Scale bars 10 μm.
DOI: https://doi.org/10.7554/eLife.39180.008

The following figure supplement is available for figure 3:

**Figure supplement 1.** β2 fusion constructs and expression levels.
DOI: https://doi.org/10.7554/eLife.39180.009

## Dissociation studies reveal two distinct binding modes for PICK1

To obtain better insight into the molecular mechanism underlying the different maximal binding levels we turned to kinetic experiments. Association experiments did not show any striking difference between LKV and LKI (*Figure 3c–d*). Both constructs displayed slow (half-bound maxima; LKV:

**Table 1.** PICK1 WT binding statistics.

| Ligand | $EC_{50}$ (nM) | | | Bmax (% of max) | N |
|---|---|---|---|---|---|
| Tac-yfp dat c24 | 47 | ± | 5 | 100 [§] | 7 |
| TAC-YFP DAT C24 + A | n.d | | | - | 3 |
| TAC-YFP GluA2 C24 | 73 | ± | 19 | 44 ± 9 [§] | 3 |
| TAC-YFP GluA2 C24 + A | n.d | | | - | 3 |
| SF-GluA2 | 67 | ± | 6 | 100 [a,nd] | 3 |
| SF GluA2 + A | n.d* | | | 33 ± 4 [a,nd] | 3 |
| β2-DAT WT (LKV) | 37 | ± | 5 | 100 [#] | 8 |
| β 2-DAT LKI | 29 | ± | 4 | 56 ± 2 [#] | 7 |
| β 2-DAT LKA | 59 | ± | 11 | 41 ± 4 [#] | 5 |
| β 2-DAT + A | 201 | ± | 102 | 21 ± 3[#] | 3 |

DOI: https://doi.org/10.7554/eLife.39180.010

24 ± 8 min, LKI: 14 ± 6 min; (means ±s.e.m., n = 3)), but saturable binding. PICK1 dissociation, on the other hand, revealed that whereas PICK dissociated very slowly when bound to LKI ($t_{1/2}$=431 ± 16 min; mean ±s.e.m., n = 3), a distinct fast component of dissociation was observed from LKV on top of the slow dissociation rate ($t_{1/2}$=21 ± 8 and 373 ± 51 min; mean ±s.e.m., n = 3), (*Figure 3e–f*). This suggests that the unbinding of LKV consists of two kinetically distinct processes and that PICK1 therefore might engage in two different binding configurations depending on the concentration of PICK1 and the intrinsic affinity for the membrane embedded ligand. That is, when the concentration of PICK1 is low and the intrinsic affinity for the membrane embedded ligand is low, PICK1 might adopt the intuitive binding mode with both PDZ domains of the dimer bound to a membrane embedded ligand (slow dissociation rate). However, when the concentration of PICK1 is high and the intrinsic affinity for the membrane embedded ligand is high, PICK1 might gradually switch to a binding mode with only one PDZ domain engaged in binding of the membrane embedded ligand (fast dissociation rate). To probe the feasibility of this hypothesis, we turned to thermodynamic simulations.

## Simulations support dual binding modes for PICK1

The principle of bivalency is well known to endow high affinity (often denoted as 'avidity') and the two individual steps in the bivalent PDZ-binding of PICK1 (aa) to two identical membrane embedded ligands (e.g. a receptor) (A) can be represented as shown in the scheme in *Figure 4a* (leading to formation of the 'red complex', aAAa). Indeed, bivalency increases the overall affinity and residence time since this permits multiple partial unbinding and rebinding events to take place before the protein fully dissociates (*Vauquelin and Charlton, 2013*; *Vauquelin and Charlton, 2013*). However, when the bulk concentration of PICK1 [aa] is sufficiently high, the free bivalent PICK1 would be expected to outpace binding of the second PDZ domain, thereby preventing formation of the 'red complex' (aAAa) and instead leading to formation of the 'blue complex' (aaAAaa) (*Figure 4a*). Under these conditions, the rate for formation of the blue 'ternary' complex ($V_2$) is larger than the rate for formation of the red complex, ($V_3$), because $V_2$ relies on the free bulk concentration of ligand [aa], whereas $V_3$ relies on the local concentration [L] of the second domain for binding, which in turn depends on the distance (r) between the individual domains (*Figure 4a*). Moreover, $V_3$ might be compromised by steric hindrance, restricted rotation freedom and entropic penalty jointly denoted, *f* (*Figure 4a*) (*Vauquelin, 2013*; *Vauquelin and Charlton, 2013*). The effect of this empiric factor is to scale the effective concentration [L] of the free end of the PICK dimer in the AAaa complex (the 'green complex'). The proposed model is the simplest model explaining our data. However, additional reaction steps can be envisioned for the binding of PICK1 for example the binding of the amphipathic helix in PICK1 to the cell membrane (*vide infra*) or putative long-range allosteric structural changes induced by the first binding event. None of such effects are necessary explicitly to describe our data, but may be encompassed by the *f* value. By simultaneously solving the system of differential equations related to formation of all of the involved complexes (*Figure 4—figure*

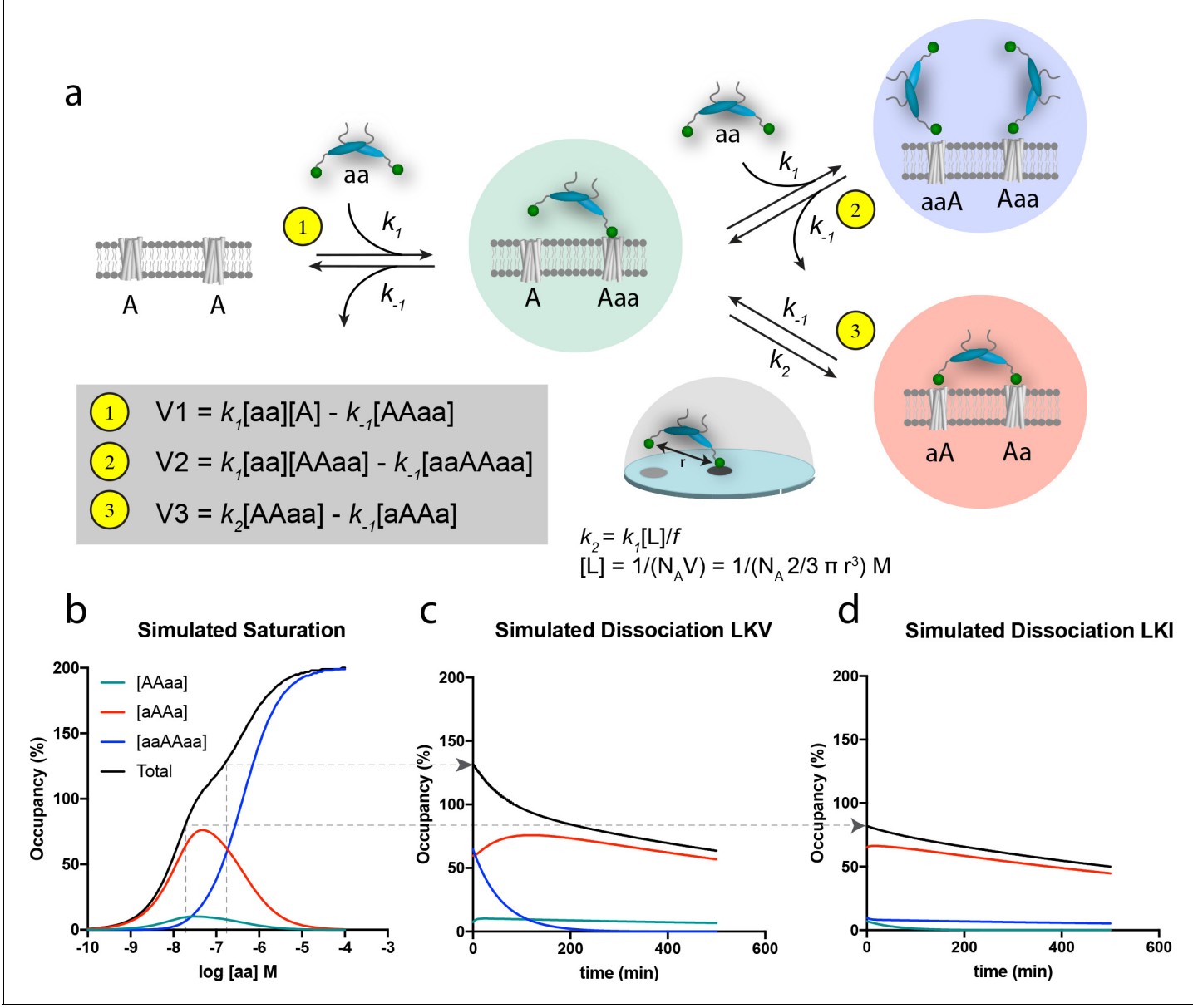

**Figure 4.** Schematic representation of the thermodynamic model for homobivalent ligand-target interactions and thereon-based simulated saturation and dissociation curves. (a) Thermodynamic scheme for homobivalent ligand, 'aa'- target, 'AA', interactions (see also *Figure 4—figure supplement 1* for full scheme). The different binding modes in panel a are designated by 'AAaa' for the partially bound complexes (green), by 'aAAa' for the bivalently bound complex (red) and by 'aaAAaa' for 'ternary' complex with two partly bound ligands (blue). The rebinding kinetics is dependent the local concentration, [L], that is calculated as that of one molecule within a half-sphere with radius, r. Moreover, the rate constant is modified by *f* due to steric hindrance, restricted rational freedom and entropic cost. An *f* of 185 enables good qualitative simulation of our data (see also *Figure 4—figure supplement 3* for behavior at other values of *f*). (b) Simulated saturation binding curve for binding of species. Input parameters: $k_1 = 1.85 \cdot 10^5$ $M^{-1}min^{-1}$, $k_{-1} = 0.0085$ min$^{-1}$, $k_2 = 0.136$ min$^{-1}$ (i.e. a composite rate constant such as defined in the figure). Total incubation time is 120 min. Analysis of the total signal according to a variable slope sigmoidal dose-response paradigm yields half-maximal signal at 50 nM. Note for these parameters the blue 'ternary' complex outpaces the red bivalent complex at bulk concentration of PICK1 above 100 nM. (c,d) Simulated dissociation curves after 120 min pre-incubation with 200 nM (c) and 20 nM (d) of the same homobivalent ligand as in panel c (corresponding to the affinity difference between LKV and LKI). [aa] is set and kept at 0 for simulating the 'washout phase'.

DOI: https://doi.org/10.7554/eLife.39180.011

The following figure supplements are available for figure 4:

**Figure supplement 1.** Differential equations to follow the time-wise changes in the different target species shown in *Figure 4*.
DOI: https://doi.org/10.7554/eLife.39180.012

**Figure supplement 2.** comparison of the PICK WT - LKV binding curves derived from experiments and the simulation.

*Figure 4 continued on next page*

*Figure 4 continued*

DOI: https://doi.org/10.7554/eLife.39180.013

**Figure supplement 3.** Broader range of *f* ratios and homobivalent ligand concentrations to illustrate the influence of this parameter on saturation profiles.

DOI: https://doi.org/10.7554/eLife.39180.014

supplement *1a–b*), and using $k_1$ and $k_{-1}$ values derived of from the in-solution PDZ binding to soluble ligands (*Erlendsson and Madsen, 2015*; *Erlendsson et al., 2014*) together with an inter PDZ distance (*r*) of 180 Å determined from the structure of PICK1 based on Small-Angle X-ray Scattering (*Karlsen et al., 2015*)) and an *f* value of 185, we obtained a biphasic saturation binding curve (*Figure 4b*) that overall was in good agreement with the experimentally derived saturation binding curve observed for PICK1 binding to LKV. Importantly, the biphasic shape results from the population of the ternary blue complex (aaAAaa) outpacing the binary red complex (aAAa) at concentrations above 100 nM of PICK1. Note that in the averaged experimental data shown in *Figure 3a and c*, the biphasic behavior of the saturation binding curve is likely masked by experimental variation, as supported by the fact that we could extract a representative data set and fit this to a biphasic curve in good agreement with the simulated curve (*Figure 4—figure supplement 2a–c*).

The modeling also rationalizes the differential binding observed for LVK and LKI (*Figure 3a–d*). The lower intrinsic affinity for LKI reflects a change in the dissociation constant $k_{-1}$, and as $k_{-1}$ is equivalent for the three rates (V1, V2 and V3), the relative partitioning into the three different complexes will be unchanged. The absolute concentration dependence, however, will be parallel shifted when comparing LKV to LKI, as illustrated for a specific concentration [aa] by the arrow in *Figure 4b*. Consequently, a ligand with lower intrinsic affinity (such a LKI) will need a correspondingly higher concentration of bulk PICK1 to populate the ternary blue complex (*Figure 4*). This likely entails that what we observe when analyzing binding for the LKI ligand, that is the reduced apparent $B_{max}$ for LKI (*Figure 3b*) most likely represents a 'plateau' before transition to the blue complex, which would be observed if we experimentally were able to use even higher concentration of PICK1. Finally, it should be noted that increasing or decreasing the *f* parameter, affecting $k_2$, would also have an important influence on the concentration-dependent formation of the blue complex (*Figure 4—figure supplement 3*).

The discrimination between the different binding modes becomes even more perceptible in the simulation of the dissociation experiments; that is when free, labeled ligand molecules are removed and/or prevented to bind (*Figure 4c,d*). The observed dissociation for LKV is first dominated by the unbinding of one of the ligands in the blue complex. Being governed by a single off-rate, $k_{-1}$, the initial component of the curve is fast, as would be expected from a monovalent binding event. Yet, since the so-obtained partially bound complexes (green) are more prone to form the bivalent red complex than to dissociate, the second component is governed by the slow dissociation of fully bound bivalent complexes (*Figure 4c*). For LKI or GluA2, which have lower intrinsic affinities (numerically corresponding to lower PICK1 concentration on *Figure 4b*), the ternary 'blue' complex is not favored and only the slow second component is observed in the simulation (*Figure 4d*). Importantly, the simulations are in very good agreement with the actual dissociation experiment (*Figure 3e,f*).

## Dual binding modes for PICK1 relies on two functional PDZ domains

The model described above and in *Figure 4* relies on two independent PDZ domain ligand-binding sites in the PICK1 dimer, and consequently it predicts that no differences in the apparent maximal binding should be observed for the different ligands if one of the PDZ domains is rendered non-functional. To experimentally test this prediction, we mutated Ala87 in the PICK1 to Leu (A87L), a mutation previously described to abolish binding of ligands to the PDZ domain (*Erlendsson et al., 2014*; *Madsen et al., 2005*). We found that PICK1 A87L bind very weakly also to SCMSs (*Figure 5—figure supplement 1*). Next, we mixed monomers (obtained in 0.1% TX-100) (*Karlsen et al., 2015*) labeled with Alexa-647 (blue, WT) and Alexa-549 (red, WT or A87L) to allow formation of PICK1 WT/WT homodimers as well as WT/A87L heterodimers (upon dilution of TX-100 concentration to 0.01%, see Materials and methods for details). TX-100 (0.1%) did not affect PDZ domain function per se (*Figure 5—figure supplement 2*). Binding of bivalent homodimeric PICK1 WT/WT was then compared to monovalent heterodimeric PICK1 WT/A87L on sheets from cells expressing either LKV or

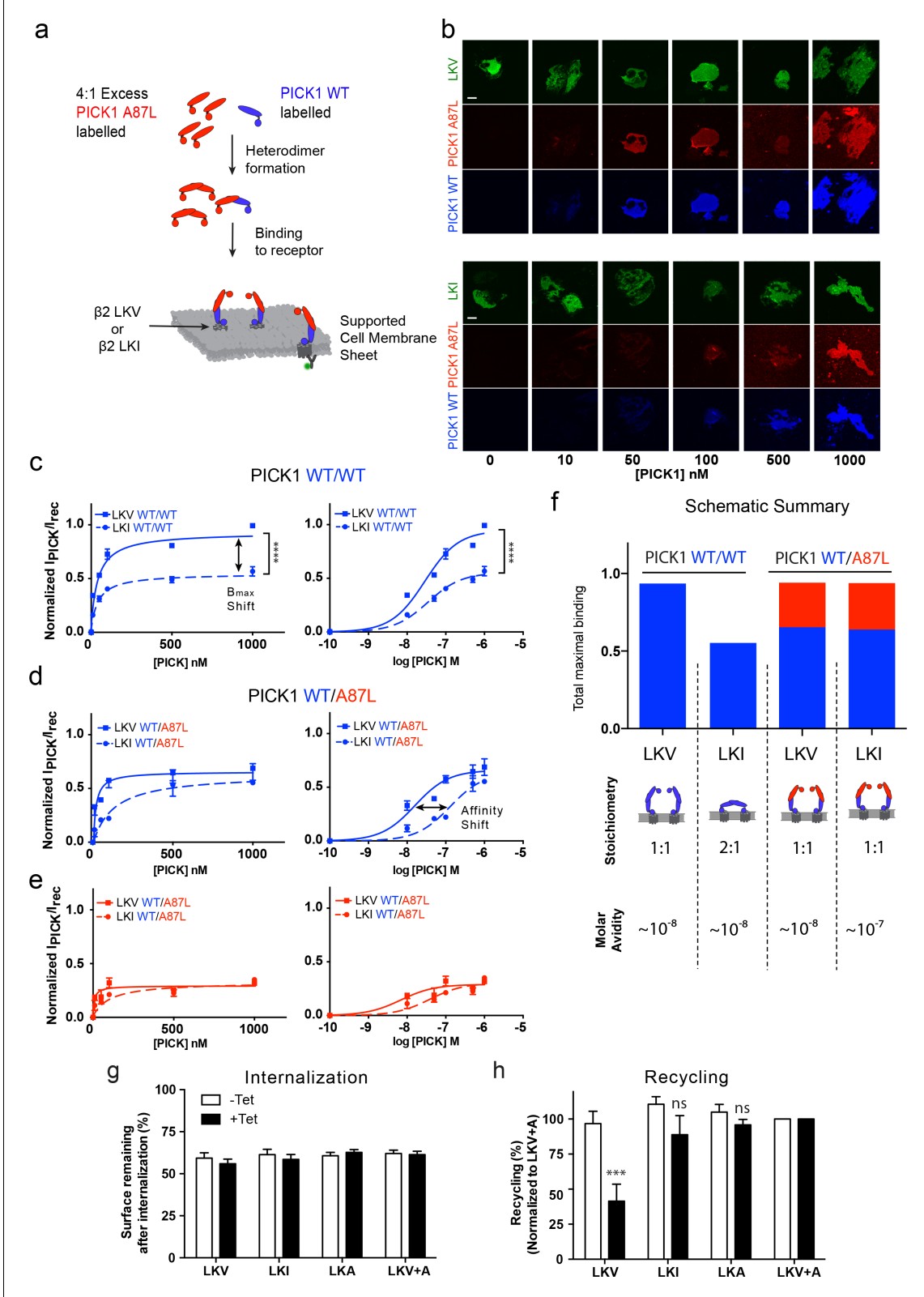

**Figure 5.** PICK1 switch scaffolding mode dependent on intrinsic PDZ affinity. (**a**) Schematic representation of the experimental setup. Both PICK1 WT (blue) and PICK1 A87L (red) are labeled. Exchange is allowed before binding of PICK1 WT/A87L heterodimers. (**b**) Representative images of heterodimeric binding as a function of concentration. (**c–e**) Normalized concentration dependent binding to either LKV (solid lines) or LKI (dashed lines), and quantified in both the WT (blue) and A87L (red) channel as function of linear and logarithmic concentration. LKV PICK1 WT/A87L:

*Figure 5 continued on next page*

*Figure 5 continued*

$K_d^*$=24 ± 21 nM, $B_{max}$: 51 ± 2%. LKI PICK1 WT/A87L $K_d^*$=123 ± 21 nM, $B_{max}$: 56 ± 2% (see *Tables 1–3* for explanation of $B_{max}$ values) (n = 3), (****p≤0.0001). (f) Schematic summary of conclusions made from figures c-e. Fitted $B_{max}$ values are shown on the y-axis, for PICK1 WT/WT and PICK1 WT/A87L to SCMS expressing LKV and LKI together with avidity and proposed binding mode (g–h) Flp-In T-REx 293 eYFP-PICK1 cells transiently expressing LKV, LKI, LKA or LKV +A with (*black bars;+Tet*) and without (*white bars; −Tet*) tetracycline-induced expression of eYFP-PICK1 were surface-labeled with anti-FLAG M1 antibody prior to stimulation of internalization with agonist (10 µM Isoproteronol for 20 min). Subsequently, cells were treated with the antagonist (10 µM Alprenolol, 60 min) to allow recycling to the plasma membrane. Surface receptor immunoreactivity was determined by surface ELISA. Internalization (g) refers to the fractional reduction of surface receptor in response to 25 min of agonist exposure compared with non-treated cells (100%). Recycling (h) refers to the fractional recovery of surface receptor following antagonist incubation for 1 hr. These values were all normalized to the respective signal of non-interacting β2-LKV + A, with and without tetracycline induction, respectively. Data are means ±S.E, n = 6. (***p<0.001, ns >0.05, One way ANOVA, Bonferroni's post test compared to LKV +A). Scale bars 10 µm.
DOI: https://doi.org/10.7554/eLife.39180.015

The following figure supplements are available for figure 5:

**Figure supplement 1.** Mutation of the PICK1 PDZ domain abolished binding to SCMSs.
DOI: https://doi.org/10.7554/eLife.39180.016
**Figure supplement 2.** Triton-X 100 does not impair PICK1 PDZ domain function.
DOI: https://doi.org/10.7554/eLife.39180.017
**Figure supplement 3.** PICK1 WT binding does not promote nucleation of PDZ independent binding on the membrane sheets.
DOI: https://doi.org/10.7554/eLife.39180.018

LKI (*Figure 5a,b*). Importantly, PICK1 WT/WT binding to LKV revealed again a two-fold higher $B_{max}$ than for LKI with no difference in binding strength (*Figure 5c*).

For binding of PICK1 WT/A87L heterodimers, we likewise obtained signals in both the PICK1 A87L (red) and the PICK1 WT (blue) channels (*Figure 5b*) confirming formation of the heteromers. No PICK1 A87L signal was observed by incubation together with PICK1 WT without allowing subunit exchange, showing that PICK1 WT does not nucleate binding of A87L for example by oligomerization (*Figure 5—figure supplement 3a,b*). In agreement with our model, we did no longer observe any difference in $B_{max}$ between LKV and LKI for the PICK1 WT/A87L heterodimers, demonstrating that the difference in $B_{max}$ indeed must rely on PDZ domain bivalency (*Figure 5d,e*). Moreover, the resulting binding curves for LKV showed a binding strength similar to that of the PICK1 WT homo-dimers (WT/WT $K_d^*$=37 ± 5 nM; WT/A87L $K_d^*$=24 ± 21 nM, mean ±s.e.m., n = 3) (*Figure 5d,e*), suggesting that one PDZ domain is sufficient to support the strong binding and corroboration of the ternary complex binding mode (blue complex in *Figure 4a*). For LKI, however, we observed a significant decrease in the apparent affinity of PICK1 WT/A87L heterodimers in both channels (WT/WT $K_d^*$=29 ± 4 nM, WT/A87L $K_d^*$=123 ± 41 nM, mean ±s.e.m., n = 3) (*Figure 5d,e*), suggesting that the high binding strength observed for WT/WT does rely on the dual PDZ domains in agreement with the model. PICK1 A87L homomers do not interfere with PICK1 WT homomer binding if exchange between monomers has not been allowed (*Figure 5—figure supplement 3c,d*).

Interestingly, by tentatively combining the binding signal of each components of the dimer (WT (blue)/A87L (red) the maximal binding for WT/A87L on LKI (and LKV) approached that of WT/WT on LKV, suggesting an overall increase in the number of binding sites of LKI when going from WT/WT to WT/A87L (*Figure 5f*).

Altogether, these findings strongly support a 'dual scaffold mode' as outlined in our model and simulations above; that is, one PICK1 WT/WT dimer preferentially binds one membrane embedded LKV ligand (i.e. with 1:1 stoichiometry). In contrast, one PICK1 WT/WT dimer preferentially binds two membrane embedded LKI ligands (i.e. with 1:2 stoichiometry), thereby preserving a high binding strength due to the avidity obtained by doubling the PDZ interactions. Upon compromising the PDZ bivalence (PICK1WT/A87L), however, PICK1 is forced to obtain the 1:1 configuration, which increase $B_{max}$ at the expense of binding strength for LKI (*Figure 5f*).

## Dual scaffold modes determine the ability of PICK1 to regulate receptor recycling

To test whether the differential PICK1 scaffold modes could have a functional consequence, we tested the ability of PICK1 to inhibit recycling of its interaction partners (*Citri et al., 2010*; *Madsen et al., 2012*). As model system we used our flagß2-DAT8 constructs expressed in HEK293

FlpIN cells with tetracycline inducible expression of eYFP-PICK1 (*Madsen et al., 2012*). Expression of eYFP-PICK1 did not affect the isoproterenol-induced internalization of any of the constructs (*Figure 5g*), however, expression significantly inhibited the reinsertion into the plasma membrane of LKV normalized to LKV +A after alprenolol treatment (41.5 ± 12% vs 100%, mean ±s.e.m., n = 6) as previously described (*Madsen et al., 2012*). In contrast, eYFP-PICK1 expression did not significantly affect the reinsertion of either LKI or LKA (88.8 ± 14% and 95.9 ± 3.9%, respectively, mean ±s.e.m., n = 6) (*Figure 5h*) despite the fact that these display similar affinities on SCMSs. This suggests that the ability of PICK1 to reduce recycling of an interaction partner could be dependent on formation of the ternary (blue) complex and hence the ability to recruit for example kinases (*Figure 4a*).

## The PICK1 amphipathic helix contributes to avidity of the binding to membrane sheets and is necessary for synaptic localization

To further investigate how one PDZ domain in the PICK1 dimer is sufficient to mediate interaction with a membrane embedded binding partner with nanomolar binding strength (LKV), we mutated two hydrophobic residues in the membrane binding amphipathic helix preceding the BAR domain in PICK1 (PICK1 V121E,L125E) (*Figure 6a*) (*Holst et al., 2013*; *Herlo et al., 2018*). PICK1 V121E,L125E showed markedly reduced binding strength for the LKV (PICK1 V121E,L125E, $K_d$*=286 ± 126 nM, mean ± s.e.m., n = 3) on SCMSs compared to PICK1 WT without affecting maximal binding (*Figure 6b* and *Table 3*). This supports that the binding strength of the PICK1 WT relies in part on membrane binding.

We also assessed the possible functional consequence of reduced binding strength of this PICK1 variant. Expression of eYFP-PICK1 V121E,L125E reduced the reinsertion of LKV significantly (+Tet 50.9 ± 8.3% vs –Tet 86.0 ± 13%, mean ±s.e.m., n = 6) and to the same extend as eYFP-PICK1 WT (+Tet 51.3 ± 3.1 vs –Tet 81.7 ± 7.7, mean ±s.e.m., n = 6) (*Figure 6c–d*), suggesting that this function of PICK1 is independent on the avidity provided by the membrane binding helix, but instead relies on a sufficiently high PICK1 concentration to achieve formation of 1:1 configuration (*Figure 4*, blue).

Finally, we tested the role of the membrane binding amphipathic helix in synaptic localization of PICK1 by employing a lentiviral molecular replacement strategy in hippocampal neurons. Using a previously characterized construct (*Citri et al., 2010*), we knocked down endogenous PICK1 with shRNA (sh18) and replaced it with either shRNA-resistant GFP-PICK1 WT or GFP-PICK1 V121E, L125E. Similarly, to endogenous PICK1, virally expressed GFP-PICK1 WT showed partial clustering and significant localization to dendritic spines, which was reflected in a high level of co-clustering with the synaptic marker PSD-95 (*Figure 6f*, colocalization shown in white in merged picture). GFP-PICK1 V121E,L125E had similar levels of expression in somatic regions (101 ± 8% of wt, p=0.94, n = 16) (*Figure 6g*), but distributed much less into the dendrites and displayed a more diffuse localization than GFP-PICK1 WT. The total number of GFP-PICK1 V121E,L125E clusters relative to PSD-95 clusters was significantly reduced compared to GFP-PICK1$^{WT}$ (48.5 ± 7% of GFP-PICK1 WT, p<0.0001, n = 24) (*Figure 6h*), whereas the number of PSD-95 clusters per length of dendrite was unchanged (97 ± 10%, p=0.81, n = 24) (*Figure 6i*). Interestingly, the localization to dendritic spines and consequently the co-clustering of GFP-PICK1 V121E,L125E with PSD-95 was almost abolished compared to GFP-PICK1 WT (0.17 ± 0.04, p<0.0001, n = 24) (*Figure 6j*), This demonstrates that the binding strength provided by the membrane binding capability of the amphipathic helix is essential for stable synaptic localization of PICK1.

## Discussion

Here, we utilize SCMS's to elucidate how PDZ scaffold proteins bind transmembrane proteins in a controlled cell membrane environment. As a result of the avidity of multiple protein-protein and/or protein-lipid interactions, the two PDZ scaffold proteins PICK1 and PSD-95 both displayed binding strength several orders of magnitude higher than the intrinsic affinities for individual PDZ interactions (see *Tables 1–3*). For PICK1, we sought to address the relative role of native constituents of the SCMSs to this increased binding strength. We probed the binding to the membrane and its native proteins constituents by either disrupting the PDZ binding sequence of the overexpressed membrane-protein (*Figure 1f*, *Figure 2 – figure supplement 1b* and *Figure 3a*), which reduced the maximal binding to 20–30% of the control depending on the construct. Conversely, disrupting the PICK1 PDZ domain (A87L) eliminated binding almost completely (*Figure 5—figure supplement 1b*).

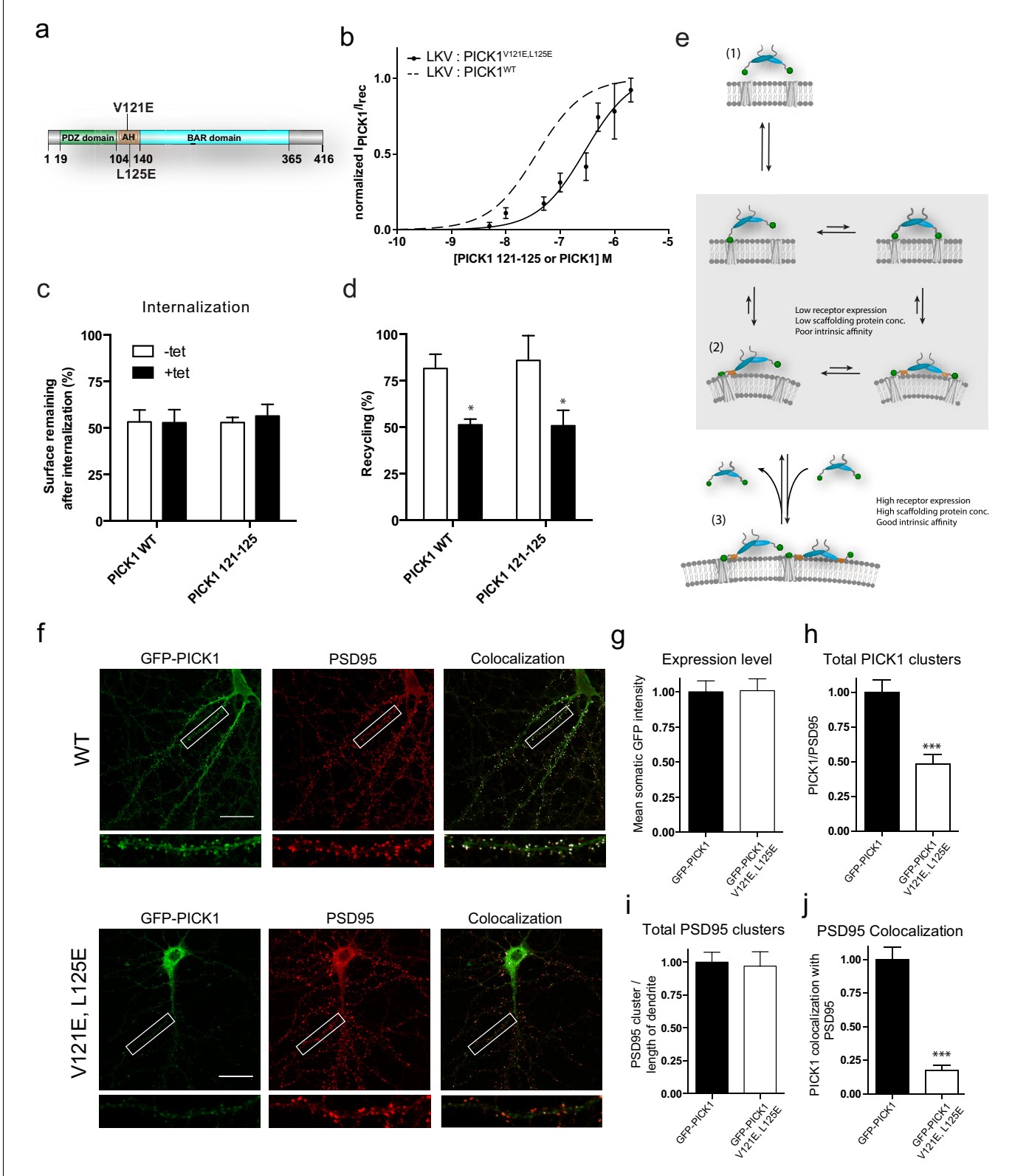

**Figure 6.** Disruption of the PICK AH compromises binding strength and alters synaptic localization and clustering of PICK1 in hippocampal neurons. (**a**) PICK1 V121E,L125E amphipathic helix mutations depicted on PICK1 domain overview. (**b**) Normalized malemide stained PICK1 V121E,L125E binding as a function of concentration towards LKV. $K_d^* = 0.28 \pm 0.12$ µM, $B_{max}$: 95 ± 9%. Malemide stained PICK1 WT as dashed line for comparison. (**c–d**) Flp-In T-REx 293 eYFP-PICK1 WT or eYFP-PICK1 V121E, L125E cells transiently expressing LKV with (*black bars;+Tet*) and without (*white bars; −Tet*) similar to
*Figure 6 continued on next page*

*Figure 6 continued*
the data shown in **Figure 3h–g**. Data are means ± S.E, n = 6. (*p≤0.05, Students t-test, correction for multiple comparisons). (**e**) Model of PICK1 scaffolding. After the initial binding using one PDZ domain, PICK1 will be able to maintain avidity via both the second PDZ domain and the AH (grey box). The AH interaction, however, is dependent on the PDZ interaction. Under conditions yielding the second interaction relatively unfavorable state (2) will be slightly favored. This will allow for an additional PICK1 molecule to bind the free receptor site. If the concentration and intrinsic affinity is sufficiently high this state (3) can be maintained. (**f**) Representative images of hippocampal neurons, DIV 20–22, expressing GFP-PICK1WT (upper panel) or GFP-PICK1 V121E, L125E (lower panel) stained for GFP (left column) and PSD-95 (middle column), and the overlay is shown in the right column. Scale bar is 20 µm. (**g**) Quantification of the expression level of GFP-PICK1 WT and GFP-PICK1 V121E, L125E measured as the mean intensity of GFP in the soma. (**h**) Quantification of the number of dendritic PICK1 clusters (see Materials and method section for definition) relative to the number of PSD-95 clusters. (**i**) Quantification of PSD-95 clusters relative to dendritic length. (**j**) Quantification of the fraction of PICK1 clusters colocalized with PSD-95. All values are normalized to the corresponding mean value for GFP-PICK1 wt. (***p<0.001).
DOI: https://doi.org/10.7554/eLife.39180.019

These findings have two implications: 1) PICK1 binds native membrane proteins in the SCMS using the PDZ domain and 2) PICK1 does not bind the membrane in absence of PDZ interactions. Given the slow non-equilibrium conditions of the system, the interplay between the overexpressed membrane proteins and the native constituents of the SCMS are not easily interpreted. Disruption of the membrane binding amphipathic helix in PICK1 reduced the affinity of the interaction (**Figure 6b**), suggesting that the interaction with the membrane does play a role in context of the PDZ binding. Conversely, we would argue that the PDZ interaction with transmembrane proteins native to the SCMS would be effectively competed by the overexpressed transmembrane proteins (given their comparable affinities for example **Figures 1f** and **3a**) and consequently play a minor role in context of overexpression.

Regardless the molecular explanation of the high binding strength in the SCMSs, it implies that often scaffold proteins may be at saturating concentrations in vivo – in particular given high local concentrations of transmembrane interaction partners. Tuning of affinities/avidities may therefore represent a relative inefficient mechanism of regulation, which we importantly could demonstrate by the ability of mutant PICK1 (PICK1[V121E,L125E]) to retain cellular function despite a ten-fold drop in binding strength according to the SCMS assay.

The slow binding kinetics observed for PICK1, on the other hand, imply that the scaffold interactions that underlie biological regulation are likely to occur under non-equilibrium conditions. Consequently, factors affecting the binding kinetics (e.g. sterical hindrance and flexibility) may have hitherto unappreciated biological impact. Similar slow kinetics were recently demonstrated for bivalent receptor ligands by kinetic simulations (**Vauquelin and Charlton, 2013**). The slow off-rates observed for PICK1 moreover provide a putative biological mechanism to convert at transient signal (such as an activity dependent exposure of the GluA2 C-terminus) into a long-lived effect (e.g. membrane localization of PICK1).

Our most striking finding was that intrinsic PICK1 PDZ affinities did not correlate directly to the binding strength determined on SCMSs, but surprisingly resulted in changes in the observed maximal binding. We initially considered, that this might involve an element of 'kinetic proofreading' (**Hopfield, 1974**; **McKeithan, 1995**) whereby the PDZ domain would enable insertion of the

**Table 3.** PICK1 Mutants binding to LKI

| PICK1 variant (LKI) | $EC_{50}$ (nM) | | | Bmax (% of max) | N |
|---|---|---|---|---|---|
| Pick1 wt | 29 | ± | 4 | 56 ± 2 [*] | 7 |
| PICK1 A87L | n.d | | | - | 3 |
| Half label PICK1 A87L/**WT** | 123 | ± | 41 | 56 ± 2 [*] | 3 |

n.d: No detectable binding or that curves could not be reliably fitted to a saturable binding event.

n.d* Curve not fitted but for comparison an apparent Bmax is determined from the concentration endpoint.

[*] Comparable Bmax values

[§] Comparable Bmax values

[&] Comparable Bmax values

DOI: https://doi.org/10.7554/eLife.39180.020

**Table 2.** PICK1 Mutants binding to LKV.

| PICK1 variants (LKV) | EC$_{50}$ (nM) | | | Bmax (% of max) | N |
|---|---|---|---|---|---|
| Pick1 wt | 37 | ± | 5 | 100 [#] | 8 |
| PICK1 A87L | n.d | | | - | 3 |
| PICK1 L121E V125E | 286 | ± | 126 | 95 ± 9 [#] | 3 |
| Half label PICK1 A87L/**WT** | 24 | ± | 21 | 51 ± 2 [#] | 3 |

DOI: https://doi.org/10.7554/eLife.39180.021

amphipathic helix (*Herlo et al., 2018*) in the membrane given sufficient residence time, however, the difference in maximal binding was preserved after mutation of the amphipathic helix. Instead, our kinetic modeling together with the experimental data suggests that this behavior reflects an affinity dependent switch in the scaffolding mode of PICK1 – that is, using both PDZ domains for interacting with membrane associated PDZ partners with low intrinsic affinity, but only one PDZ domain when the local bulk concentration is above or close to the intrinsic affinity. This behavior was recently predicted for bivalent ligands too using kinetic simulations (*Vauquelin, 2013*; *Vauquelin and Charlton, 2013*). Thus, after binding to the first binding site penalties associated with binding of the second site of the bivalent interaction (e.g. from reduced entropy, straining of the molecule and steric hindrance) may rather favor binding of a second molecule from the solution (*Figure 4a*). Importantly, this behavior is likely favored by the large distance (~20 nm) between the two PDZ domains in PICK1 (*Karlsen et al., 2015*), which will render the effective local concentration, [L], of the second PDZ domain after binding of the first relatively low compared to for example the tandem domains in PSD95. On the other hand, one might expect the steric restriction, *f*, of the two closely spaced and structurally aligned PDZ grooves in PSD95 PDZ1 and 2 to be more prominent than in the flexible PICK1. We also considered modeling explicitly an allosteric effect in the binding of PICK1 to the membrane embedded ligands (*Kramer and Karpen, 1998*; *Lu and Ziff, 2005*). However, the evidence for allostery in PICK1 is controversial (*Karlsen et al., 2015*) and an allosteric model is not needed to explain our data. Still, it should be noted that allostery could implicitly be contained in the *f* value. To better understand the complex behavior between scaffold proteins and proteins embedded in biological membranes further structural and dynamic insight of the system from experiments and molecular dynamics simulations would be needed combined with for example a statistical mechanics approach.

Functionally, the scaffolding mode for high intrinsic affinity ligands (including DAT and NET) (*Erlendsson et al., 2014*) would leave one PDZ domain free for recruitment of cytosolic proteins (e.g. kinases, phosphatases, GTPases). Only this mode supports the PICK1 function in regulating recycling of the β2AR. Since fast recycling of β2AR has been shown to rely on the PKA phosphorylation of S345/S346 (*Vistein and Puthenveedu, 2013*), and the PICK1 PDZ domain binds both Calcineurin B (*Iida et al., 2008*) and PKA regulatory subunits (Ammendrup-Johnsen, Gether, Madsen, Unpublished results), PICK1 might regulate the β2-DAT trafficking by recruiting either of these components.

For ligands with lower intrinsic affinity (including GluA2, ASIC1a, HER2, Glt1b, mGluR7b and Ephrin B1) (*Erlendsson et al., 2014*), however, both PDZ domains would be engaged with membrane-associated ligands. This scaffold mode will potentially lead to clustering of the ligands and possibly changes in lateral diffusion, rotational flexibility, and molecular orientation - including positioning of the C-terminus relative to the membrane. Posttranslational modification of the C-termini or PICK1 as well as variations in the local concentration of PICK1 might complicate this simplified scheme.

Finally, since the PICK1 N-BAR domain, like other N-BAR proteins, is recruited to biological membranes in a curvature sensitive manner (*Herlo et al., 2018*) the PICK1 concentration on membranes might be almost two orders of magnitude higher in areas of high curvature such as endocytic structures, endosomes or vesicles in the biosynthetic pathway (*Bhatia et al., 2009*). This, in turn, would strongly drive interaction with membrane embedded interaction partners towards the 'blue' ternary configuration with one PDZ domain open for recruitment of cytosolic proteins thereby evoking spatio-temporal control on such proteins. This may indeed be the actual scenario for the modulation of the recycling of the β2-DAT constructs.

Vice versa, the switch in scaffolding mode would allow stoichiometrically more binding to membrane embedded ligands in areas of membrane curvature possibly feeding into the already recursive nature of membrane curvature sensing and deformation (*Madsen and Herlo, 2017*). Consequently, it would be very interesting to address, whether the clustered nature of the PICK1 binding on the SCMS's would coincide with areas of high membrane curvature.

In summary, we present a novel experimental angle on scaffold processes, allowing quantitative description of the ensemble of protein-protein as well as protein-lipid interactions on a cell membrane and our results redefine several aspects of the scaffold function. We firmly believe that this approach will continue to shape our understanding of the protein ensembles that orchestrate cellular signaling and trafficking processes with the aim of developing rational therapeutic strategies for targeting these interactions in disease.

# Materials and methods

**Key resources table**

| Reagent type (species) or resource | Designation | Source or reference | Identifiers | Additional information |
|---|---|---|---|---|
| Gene (*Rattus Norvegicus*) | pick1 | | UNIPROT - Q9EP80 | |
| Gene (*Rattus Norvegicus*) | gluA2 | | UNIPROT - P19491 | |
| Gene (*Rattus Norvegicus*) | B2AR | | UNIPROT - P10608 | |
| Gene (*Rattus Norvegicus*) | B1AR | | UNIPROT - P18090 | |
| Gene (*Rattus Norvegicus*) | PSD95 | | UNIPROT - P31016 | |
| Recombinant DNA reagent | FLAG-$\beta_2$ DAT8 LKV | *Madsen et al., 2012* | See Materials and methods Section | FLAG-$\beta$2ARHis6 Provided by Dr. Mark von Zastrow , vector: pcDNA3.1, |
| Recombinant DNA reagent | FLAG-$\beta_2$ DAT8 LKI | This paper | See Materials and methods Section | |
| Recombinant DNA reagent | FLAG-$\beta_2$ DAT8 LKA | This paper | See Materials and methods Section | |
| Recombinant DNA reagent | FLAG-$\beta_2$ DAT8 LKV + A | *Madsen et al., 2012* | See Materials and methods Section | |
| Recombinant DNA reagent | FLAG-$\beta$1AR | | | |
| Recombinant DNA reagent | SF-GluA2 | This paper | See Materials and methods Section | pRK5 pHluorin-GluA2 was kindly provided by Dr Richard Huganir. Vector pRK5 |
| Recombinant DNA reagent | SF-GluA2 + A | This paper | See Materials and methods Section | Vector pRK5 |
| Recombinant DNA reagent | Tac-YFP-GluA2 | This paper | See Materials and methods Section | pcDNA3.1 |
| Recombinant DNA reagent | Tac-YFP-GluA2+A | This paper | See Materials and methods Section | pcDNA3.1 |
| Recombinant DNA reagent | Tac-YFP-DAT | This paper | See Materials and methods Section | pcDNA3.1 |
| Recombinant DNA reagent | Tac-YFP-DAT+A | This paper | See Materials and methods Section | pcDNA3.1 |
| Recombinant DNA reagent | PICK1 WT | *Madsen et al., 2005* | See Materials and methods Section | pET41a |

*Continued on next page*

*Continued*

| Reagent type (species) or resource | Designation | Source or reference | Identifiers | Additional information |
|---|---|---|---|---|
| Recombinant DNA reagent | PICK1 A87L | *Madsen et al., 2005* | See Materials and methods Section | pET41a |
| Recombinant DNA reagent | PICK1 L121E, V125E | *Herlo et al., 2018* | See Materials and methods Section | pET41a |
| Recombinant DNA reagent | SNAP-PICK1 | This paper | See Materials and methods Section | pET41a |
| Recombinant DNA reagent | SNAP-PICK1 A87L | This paper | See Materials and methods Section | pET41a |
| Recombinant DNA reagent | PSD95 PDZ 1–2 | | | Provided by Dr. Kristian Strømgaard |
| Recombinant DNA reagent | GFP PICK1 WT | *Herlo et al., 2018* | See Materials and methods Section | pcDNA5/FRT/TO |
| Recombinant DNA reagent | GFP PICK1 V121E, L125E | *Herlo et al., 2018* | See Materials and methods Section | pcDNA5/FRT/TO |
| Recombinant DNA reagent | pOG44 vector | V600520 (ThermoFisher) | | |
| Cell line (Human) | HEK293 GripTite | Available from Thermofisher: A14150 | RRID:CVCL_U428 | Provided by Dr Jonathan Javitch. Vector pRK5 |
| Cell line (Human) | Flp-In T-REx 293 PICK1 | R78007 (ThermoFisher); *Madsen et al., 2012* | RRID:CVCL_U427 | |
| Cell line (Human) | Flp-In T-REx 293 PICK1 | R78007 (ThermoFisher); This paper | | |
| Antibody | ANTI-FLAG M1 | F3040 (Sigma) | RRID:AB_439712 | (1:1000) |
| Antibody | horseradish peroxidase-conjugated goat anti-mouse IgG | (ThermoFisher) | RRID:AB_2535981 | (1:500) |
| Peptide, recombinant protein | OrG DAT C13 | Synthesized (TAG-Copenhagen) | | |
| Chemical compound, drug | alprenolol | A0360000 (Sigma) | | |
| Chemical compound, drug | isoproterenol | 1351005 (Sigma) | | |
| Chemical compound, drug | poly-L-ornithine hydrobromide | P-8638 (Sigma) | | |
| Other | Alexa Fluor C5 malemide 488/568 | A10254, A20341 (ThermoFisher) | | |
| Other | SNAP-surface dye 568/647 | S9112S, S9137 (New England Biolabs) | | |
| Other | Alexa Fluor 488 (NHS) | A20000 (ThermoFisher) | | |
| Other | SuperSignal ELISA Femto Substrate | 37075 (Thermo Fisher) | | |
| Software, algorithm | Graphpad Prism | | | |
| Software, algorithm | ImageJ | | | |
| Software, algorithm | Copasi | | | |

## Molecular biology

pRK5 SF-GluA2 was constructed by PCR amplifying the DNA sequence encoding the signal flag sequence (SF) from a previous SF-β(2)-adrenergic receptor construct (*Madsen et al., 2012*) with sequences coding for the restriction sites XhoI and AgeI attached in the ends of the primers. Using these restriction enzymes the PCR product was inserted into a pRK5 pHluorin-GluA2 construct (a gift from Richard Huganir, Baltimore) thereby removing the sequence coding for the pHluorin in the process. In the pRK5 SF-GluA2 +A construct an alanine was added in the end of the construct by a one step quick change. pcDNA3.1 SF-β(2)-adrenergic receptor constructs LKI and LKA were made from the previous pcDNA3.1 SF-β(2)-adrenergic receptor-DAT-LKV (*Madsen et al., 2012*) using one step quick changes. pcDNA3.1 Tac-YFP-DAT/GluA2 constructs were constructed by PCR amplifying the sequence encoding YFP from a pEYFP-C1 vector and inserting this in a previous TAC-DAT construct (*Madsen et al., 2012*) using a single HindIII site between the TAC and YFP sequence. The generation of FLAG-$\beta_2$ DAT LKV, with the 8 C-terminal residues of the human DAT (-TLRHWLKV) and LKV + A with the 8 C-terminal residues of the human DAT with an additional alanine that disrupts the PDZ binding to PICK1 (*Bjerggaard et al., 2004*) (-TLRHWLKVA), was described previously (*Madsen et al., 2012*). FLAG-$\beta_2$ DAT LKI and LKA were generated similar by PCR and the resulting fragments were cleaved with KpnI and BamHI and ligated into pcDNA3 FLAG-$\beta_2$ARHis$_6$ (kind gifts from Dr. Mark von Zastrow, Departments of Psychiatry and Cellular and Molecular Pharmacology, University of California, San Francisco, CA). The V121E, L125E double mutation was introduced into pcDNA5/FRT/TO eYFP-PICK1 (rat) by quick change PCR and used for generation of a tetracyclin inducible stable Flp-In T-REx 293 cell line (see below). The V121E, L125E double mutation was subsequently subcloned into FUGWH1sh18GFPPICK1 WT (kind gift from Dr. Malenka, using a BsrG1 fragment and checking for orientation creating FUGWH1sh18GFPPICK1 V121E, L125E. SNAP-PICK1 was made by sub-cloning SNAP into the single MfeI site between the thrombin cleavage site and PICK1 in pET41 PICK1 (*Madsen et al., 2005*) resulting in an N-terminal fusion of PICK1. PICK1 A87L was introduced in pET41a SNAP-PICK1 by quick-change PCR. PICK1 L121E, V125E was sub-cloned from the FUGWH1 vector into a pET41a vector reversing the R411 to the native G411. PSD95 PDZ1-2 was a kind gift from Kristian Strømgaard and was prepared ultimately as described in *Bach et al. (2008)*.

## Protein expression and purification

SNAP-PICK1 WT, SNAP-PICK1 A87L, PICK1 WT, PICK1 A87L and PICK1 121–125 proteins are all expressed in E.Coli BL21 DE3 pLysS in the pET41 vector in LB medium. PSD-95 PDZ1-2 were expressed in *E. coli* One Shot BL21 Star (DE3). Induced using 75 mg/L IPTG and grown at 30°C for 4 hr. For purification the cells were lysed and centrifuged for 30 min at 18000 rpm. Subsequently glutathione sepharose 4B beads (Life technologies) were added to the supernatant and the suspension was incubated under slow rotation for 1 hr at 4°C. The beads were pelleted (5 min 3000 g at 4°C) and washed in TBS buffer (50 mM Tris pH 7.4, 125 mM NaCl, 0.01% or 0.1% TX-100, 1 mM DTT (no DTT if malemide staining)). The beads were then transferred to PD-10 spin columns (BIO-RAD). For SNAP staining, SNAP-surface 488,549,647 was used (NEB). Labeling efficiency for PICK1 was 72 ± 24% (average ±SD across all experiments). For maleimide staining PICK1 WT and PICK1 L121E, V125E Alexa C5 malemide 488, 568, 647 (ThermoFisher) was used. The beads were then washed thoroughly to remove excess dye and then cleaved from the beads using Thrombin (0.075 U/µl Novagen). Prior to use protein was ultra-centrifuged at 100,000 g for 30 min to remove aggregates.

The PSD95 PDZ1-2 domain protein was labeled with Alexa Fluor 488 Carboxylic succimidylester (NHS) (ThermoFisher) The buffer of the purified protein was changed to a 0.1 M sodium bicarbonate solution. 50–100 µL of the reactive dye (1 mg/ml) was added under constant stirring. The mixture was incubated for 1 hr at room temperature before the reaction was stopped by the addition of 100 mM Tris buffer (pH 8.3) and incubation for 15 min. Labeled protein and free dye were separated with a Micro Bio-Spin column P-30, and another buffer exchange was performed to get the protein in HEPES buffer (10 mM HEPES, 150 mM NaCl, pH 7.4).

Before use, protein concentration and degree of labelling (DOL) of all constructs were measured taking both the protein and the dye into account

$$DOL = \frac{A_{dye} \cdot \varepsilon_{280}}{\left(A_{280} - A_{dye} \cdot \frac{\varepsilon_{280}}{\varepsilon_{dye}}\right) \cdot \varepsilon_{dye}}$$

where $A_{280}$ and $A_{dye}$ represent the sample's absorbance at 280 nm and the dye excitation wavelength, respectively, $\varepsilon_{280}$ and $\varepsilon_{dye}$ the extinction coefficients of the protein at 280 nm and of the dye, respectively.

### Cell cultures, transfection and labeling for supported cells membrane sheets

HEK 293 Grip Tite cells (kind gift from Jonathan Javitch, Columbia University, USA) grown in standard DMEM 1965 with serum and pen-strep. and 500 µg/ml geneticin. Cells were tested negative for mycoplasma. Cells were transfected using lipofectamine 2000 and 3 µg plasmid DNA. The cells were then seeded in six well plates with approx. 400.000 cells pr. well in 2 ml DMEM 1965. Cells were used no later than 24 hr later in order to avoid cell clusters. Receptor labeling was achieved by using 1 µg/ml of Alexa Fluor 488/568/647 (ThermoFisher) primary conjugated monoclonal ANTI-FLAG M1 antibody (sigma) one hour prior to use. Labelling degree is determined as above.

### Cell cultures and transfections for surface ELISA

To generate a cell line with stable tetracycline-inducible expression of YFP-tagged PICK1 V121E, L125E, we used the Flp-In T-REx system and the Flp-In T-REx 293 cell line (Invitrogen) as previously described for YFP-tagged PICK1 (*Madsen et al., 2012*). The cells were maintained in DMEM 1965 with Glutamax (l-alanyl-l-glutamine) containing 10% fetal calf serum at 37°C in a humidified 5% $CO_2$ atmosphere. Cells were tested negative for mycoplasma. Before transfection, cells were selected using 15 µg/ml blasticidin and 100 µg/ml Zeocin (both from Invitrogen). Cells (90% confluent) were transfected using Lipofectamine 2000 (Invitrogen) with a total of 3 µg of DNA in a 1:9 ratio of the pcDNA5/FRT/TO with the eYFP-PICK1 V121E, L125E insert and pOG44 vector (Invitrogen) in Opti-MEM (Invitrogen) overnight. Cells were then split to 50% confluence and grown for an additional 24 hr with no antibiotics before selection was induced using 15 µg/ml blasticidin and 150 µg/ml hygromycin. Cells were maintained until visible foci appeared after which the cells were harvested, pooled, and further maintained as a polyclonal cell line (Flp-In T-REx 293 eYFP-PICK1 V121E, L125E). For transient transfections of Flp-In T-REx 293 eYFP-PICK1 cells with B2 DAT constructs, cells were seeded in 25 cm² cell flasks ($1 \times 10^6$ cells) or 75 cm² cell flasks ($3 \times 10^6$ cells) and grown in medium without selection for ~20 hr to reach ~70% confluence. Cells were transfected with Lipofectamine 2000 (Invitrogen) for 16 hr in medium using 0.1 µg of DNA/75 cm² flask. In general, we transfected >80% of the cells. The transfected cells were trypsinized and seeded on polyornithine-coated coverslips in 6-well plates (300,000 cells/well) or 96-well ELISA plates (35,000 cells/well) for 48–72 hr prior to experiments. After ~20 hr, medium was changed to new medium without or with tetracycline (1 µg/ml) to induce expression of eYFP-PICK1.

### Preparation of supported cell-membrane sheets

Supported Cell-membrane sheets were prepared following procedure presented by *Perez et al.* (*Perez et al., 2006a*) In brief, cover glasses (Round 25 mm, thickness #1, VWR 631–1346) were clenched for 20 min at max. power in a Harric plasma cleaner and then coated in 0.3 mM poly-L-ornithine hydrobromide (Sigma-Aldrich Product # P-8638) for approx. 30 min and then washed in water. Cells were allowed to swell in ddH2O for at total of 60 s before pressing a cover glass slide with poly-L-ornithine treated surface down towards cells. The supported cell membranes on the cover glass were then overlaid with sheet buffer (120 mM KCl, 2 mM MgCl2, 0.1 mM CaCl2, 10 mM HEPES and 30 mM glucose pH 7.35) containing 1 mg/ml BSA and kept on ice for 20 min. washed and replaced with protein containing sheet buffer and incubated 2 hr on ice in the dark. Temperature was kept low to avoid aggregation of PICK1. Sheets were then washed in sheet buffer and PBS and then fixated for 40 min. using 4% PFA before mounting onto object glasses using Prolong gold anti-fade reagent (Life technologies).

## Confocal imaging

The stained sheets were visualized using a Zeiss LSM 510 confocal laser-scanning microscope using an oil immersion numerical aperture 1.4 63x objective (Zeiss, Jena, Germany). The Alexa Fluor 488 dye antibody and YFP were excited with the 488 nm laser line from an argon–krypton laser, and the emitted light was detected using a 505–550 nm bandpass filter. BG-547 SNAP was excited at 543 nm with a helium–neon laser, and the emitted light was detected using a 560–615 nm band pass filter. The Alexa Fluor 647 was excited at 633 nm with another helium-neon laser, and the emitted light was detected using a 650 nm long pass filter. Resulting images were analyzed using ImageJ software. All scale bars on shown images corresponds to 10 μm.

## SCMS saturation binding

Freshly purified SNAP-PICK1 labeled with malemide Alexa 488, 568 or 647, or SNAP surface dye, was ultra-centrifuged at *100000 g* in a Beckman airfuge to remove possible aggregates. From these a concentration series were made in intracellular mimicking buffer (see above), and the prepared supported membrane sheets were then overlaid with protein containing solution for 2 hours. The level of SNAP-PICK1 binding pr. Ligand molecule in the membrane was quantified by dividing the measured 585/633 nm SNAP-PICK1 signal by the measured 505-550 nm YFP/Alexa 488 signal. The average intensities in areas containing intact membrane sheets were recorded by manually defining regions of interest covering these. One ROI per cell from approximately 10 cells were imaged for each condition. In the subsequent averaging process the following selection criteria were applied: All images were qualitatively assessed, and SCMS with too high (oversaturated pixels) or too low (below 2σ, compared to background) were discarded. The intensities were subsequently corrected based on degree of labeling and laser gain settings. Correction of measured channel intensities:$I_{corr} = I_{obs} \cdot L_{\%} \cdot G(V) - I_{back} \cdot G(V)$. Where, $I_{obs}$ is mean intensity of sheet or ligand, $I_{back}$ is mean intensity of background. Labeling degree correction was made using $L_{\%} = \frac{I_{obs}}{labelling\ degree} \cdot [100, 1000]$. If PMT gain is adjusted this is corrected by $G(V) = \left(\frac{V_2}{V_1}\right)^{\alpha \cdot n}$, where $\alpha$ is Conductance of dynodes of PMT and $n$ the number of dynodes inside the PMT. The fractional binding is calculated using as $\frac{I_{corr}(ligand)}{I_{corr}(receptor)}$. All curves form independent measurements are combined and subsequently normalized. All binding curves are systematically performed in parallel, and in presence of positive and negative controls. The resulting sigmoidal binding curves are fitted using the relationship:

$$\frac{I_{PICK}}{I_{Rec}} = Bottom_{\frac{I_{PICK}}{I_{Rec}}} + \frac{Top_{\frac{I_{PICK}}{I_{Rec}}} - Bottom_{\frac{I_{PICK}}{I_{Rec}}}}{1 + 10^{log\frac{K_{d}*}{[PICK1]}}}$$

## SCMS competition binding

Competition binding was performed by premixing 100 nM of labeled SNAP-PICK1 with increasing concentrations of unlabeled PICK1 WT. The reaction is then allowed to equilibrate for 120 min. Because of the assumed constructed binding kinetics the resulting binding curves are fitted to the equation given above for the saturation curves, and the half maximal binding is reported as the $K_i^*$value.

## SCMS binding of PICK1 heterodimers

The heterodimers of PICK1 WT and the PDZ binding deficient mutant A87L were made by separately purification of either construct in 0.1% TX-100. After centrifugation to remove aggregates the monomeric constructs are mixed (simultaneously) in intracellular mimicking buffer where the TX-100 concentration was diluted at least 100-times at all concentrations thereby favoring formation of heterodimers. PICK1 A87L is added in 4 times excess at all PICK1 WT concentration. For the homodimer oligomerization experiments TX-100 concentration is kept at 0.01% during purification. Triton X-100 did not directly affect the intrinsic affinity of the PDZ interaction in solution. Also, under conditions that did not allow monomer exchange, we observed no significant binding of SNAP-PICK1[A87L] or effect on binding of SNAP-PICK1[WT] (*Figure 5—figure supplement 3*).

## Fluorescence polarization binding

The fluorescence polarization assay was performed essentially as described in *Madsen et al. (2005)* and *Erlendsson et al. (2014)*. For saturation binding experiments, full-length PICK1 or mutants was diluted in buffer (50 mM Tris, pH 7.4, 125 mM NaCl, 1 mM DTT, 0.01% Triton X-100) to various concentrations and a final volume of 100 l in black low-binding 96-well microtiter plates (Corning Glass). A volume of 5 µl of Oregon Green-labeled DAT C13 peptide (OrG DAT C13) was added to each well to a final concentration of 20 nM. For the FP competition binding assay, increasing concentrations of unlabeled peptide was diluted in the wells, and a fixed 70% saturating concentration of PICK1 or mutants was added together with OrG DAT C13 as described above. All plates were incubated on ice for 30 min and analyzed on a PolarStar Omega FP reader (BMG, Germany) using a 488 nm excitation filter and a 520 nm emission filter. FP is calculated using $FP = (I_V - g \cdot I_H)/(I_V + g \cdot I_H)$. Where g is the g-factor, and $I_v$ and $I_H$ are the intensities of the emission measured in the vertical and horizontal planes, respectively. For the Saturation binding the FP value is a weighted average of bound and unbound ligand and therefore the $k_d$ can be fitted using the equation: $FP - FP_f = (FP_b - FP_f) \cdot \frac{[R_t]}{k_d + [R_t]}$, where FP is the observed FP value, $FP_f$ and $FP_b$ are the FP value for free and bound ligand, respectively, and $[R_t]$ is the total PICK1 concentration. For competition binding, $ki$ values for the peptide ligands were determined by fitting the binding curves to the equation, $FP = FP_f + \frac{(FP_b - FP_f) \cdot [R_t]}{k_d \cdot \left(1 + \frac{x}{k_i}\right) + [R_t]}$,

where $x$ is the competitor and $k_d$ is the apparent dissociation constant for OrG DAT C13. The apparent *Kd* was obtained as described before from the saturation binding experiments. All peptides were purchased from Shafer, Copenhagen, Denmark.

## Receptor recycling by surface ELISA

For ELISA-based trafficking experiments, FLAG-tagged β2-AR receptor variants were labeled with 1 µg/ml M1 mouse anti-FLAG antibody for 30 min at 4°C in parallel in two 96-well plates. In half of the wells on each plate, receptor internalization was stimulated with isoproterenol (10 µM) at 37°C for 25 min. The wells on the other half of the plate were left at 37°C and are referred to as non-treated. Subsequently, the action of the internalizing agent (isoproterenol) was terminated by addition of alprenolol (10 µM). One plate was left at 4°C for 1 hr to stop further trafficking, and the other plate was left at 37°C to allow further trafficking. Subsequently, cells were washed in DMEM 1965, fixed for 10 min. at 4°C, and washed twice in PBS before 30 min blocking in PBS + 5% goat serum and incubation with 0.5 µg/ml horseradish peroxidase-conjugated goat anti-mouse IgG (Thermo Scientific). Finally, cells were washed twice in PBS + 5% goat serum and twice in PBS before addition of SuperSignal ELISA Femto Maximum Sensitivity Substrate (Thermo Scientific). The luminescence was detected in a Wallac Victor2 plate reader after 2 min. Internalization is expressed as the ratio of the surface signal from isoproterenol treated receptors relative to the non-treated cells on the 4°C plate. Recycling is expressed as the proportion of internalized receptor that was recovered at the cell surface during 1 hr. In *Figure 5h*, these values were all normalized to the respective signal of β2-LKV + A with and without tetracycline induction, respectively, which was assayed on separate plates. Statistical significance was determined using ANOVA (*Figure 5*) or Student's *t* test (*Figure 6*), multiple samples, as indicated in legend.

## Homobivalent ligand binding model and simulations

A thermodynamic cycle model provides a detailed representation of bivalent ligand binding (*Kramer and Karpen, 1998*; *Müller et al., 1998*; *Vauquelin and Charlton, 2013*) and is here adapted for homobivalent ligands, 'aa'. (*Figure 4a* and *Figure 4—figure supplement 1*). The ligand's identical binding domains, 'a', can bind simultaneously to the proximate, identical target sites 'A' (target pairs are denoted as 'AA'). Each step is a reversible bimolecular process. The initial binding event allows the second, still free binding domain to acquire a high, constant concentration, [L], near its target site. Its association to be represented by a composite first-order rate constant, $k_2$, that takes account of [L], the intrinsic association rate constant, $k_1$, and a penalty factor, '*f*' (see *Figure 4*). The model also allows a second ligand to bind to partly occupied AA. Ligands are assumed to be in large excess over the targets, so that their concentration in the bulk of the aqueous phase

remains constant over time. To simulate the concentration and/or time- wise changes in the different modes of AA occupancy (abbreviations in *Figure 4*), the corresponding differential equations (*Figure 4—figure supplement 1*) adapted from *Vauquelin (2013)* are successively solved in parallel over very small time intervals.

## Hippocampal neuronal cultures

Hippocampal neurons were prepared from embryonic day 19 Wistar rats. The hippocampi were isolated in ice-cold dissection media (HBSS (Gibco 14185–0529, 1% penStrep (Gibco 15140122), 1% pyruvate (Gibco 11360070), 1% HEPES (Gibco 15630080), 30 mM glucose) and cleared of blood vessels and meninges. The hippocampi were kept in 2 mL ice-cold dissection media until tituration. For tituration 40 μl of papain (Worthington LS003119) was added and incubated for 20 min at 37°C. The dissection media was removed, and hippocampi washed twice with culturing media (Neurobasal (Gibco 21103–049), 5% FCS, 1% PenStrep, 2% Glutamax 1 (Gibco 35050–038), 2% B27 supplement (Gibco 17504044)). Titration was performed by pipetting 10 times up and down with the a glass pipette rounded in the end by burning, followed by 10 times with a pipette with end closed to half diameter by burning. Media was added to a total of 5 mL and filtered through a 70 μm cell filter. The neurons were plated on glass slides (size 25 mm, thickness No.1, Glaswarenfabrik Karl Hecht, Sondheim, Germany) in 2 ml of culturing media with a density of 150,000 cells pr. well. Prior to culturing the glass slides were treated with concentrated nitric acid for 3–4 hr, washed in milliQ water overnight and then burned in 96% ethanol. At 1 DIV the media is replaced with culturing media without FCS.

## Transduction of hippocampal neurons with lentivirus

At 14 DIV 5 μl of lentivirus packed with FUGWH1sh18GFPPICK1 WT or FUGWH1- sh18eGFPPICK1 V121EL125E was added to each well. The lentiviral was produced as previously described (*Eriksen et al., 2009*). At 20–22 DIV the neurons were fixed in 4% PFA for 20 min (10 min on ice and 10 min at room temperature), washed three times in PBS, permeabilised and blocked in 0.05% triton X100% and 5% goat serum for 20 min at room temperature. Then labelled with mouse anti-PSD95 (Antibodies inc.#72–028) 1:500 and rabbit anti-GFP (abcam #ab290) 1:500 for 1 hr at room temperature. Followed by staining with goat-anti mouse Alexa-488 and goat-anti rabbit Alexa-568 both 1:500. After three washes with PBS and one wash with milliQ water the slides were mounted to coverslips.

## Confocal imaging of hippocampal neurons

Neurons were imaged using an inverted confocal laser-scanning microscope (LSM 510, Carl Zeiss). The Alexa 488 was excited with the 488 nm laser line from an argon–krypton laser, and the emitted light was detected using a 505–550 nm bandpass filter, whereas the Alexa 568 was excited at 543 nm with a helium–neon laser, and the emitted light was detected using a 585 nm long-pass filter. A Zeiss Plan-Neofluar 63X/1.3 oil immersion lens was used for imaging. Images were analysed with the ImageJ software. A region of interest that contains transduced dendrites after the first branch point from soma was chosen. In the PICK1 channel the image was thresholded at two times the mean intensity, in the PSD95 channel the images were threshold at a fixed value within one experiment and the number of particles quantified.

## Statistics

All statistics performed using One-way ANOVA using Dunnett's multiple comparisons post test or parametric students t-test, both using 95% confidence intervals. ns $p>0.05$, *$p\leq0.05$, **$p\leq0.01$, ***$p\leq0.001$, ****$p\leq0.0001$. All in-text numbers are reported as the mean of independent experiments $\pm$ s.e.m unless stated otherwise. Graphpad Prism 6 (La Jolla California USA, www.graphpad.com) was used for all graphs and statistics.

## Acknowledgements

We thank Donny Czerny and Nabeela Khadim for excellent technical assistance. The work was supported by the National Institute of Health Grants P01 DA 12408 (UG), the Danish Council for

independent Research – Medical Sciences (UG, KLM), University of Copenhagen BioScaRT Program of Excellence (UG, KM), the Lundbeck Foundation Center for Biomembranes in Nanomedicine (UG, KM), the UNIK Center for Synthetic Biology (UG, KLM) and the Novo Nordisk Foundation (UG)

## Additional information

### Funding

| Funder | Grant reference number | Author |
|---|---|---|
| National Institutes of Health | P01 DA 12408 | Ulrik Gether |
| Lundbeckfonden | Center for Biomembranes in Nanomedicine | Ulrik Gether Kenneth Lindegaard Madsen |
| Novo Nordisk | | Ulrik Gether |
| Det Frie Forskningsråd | | Ulrik Gether |
| Danish Council for Independent Research, Medical Sciences | | Ulrik Gether Kenneth Lindegaard Madsen |
| University of Copenhagen | BioScaRT Programof Excellence | Ulrik Gether Kenneth Lindegaard Madsen |
| University of Copenhagen | UNIK Center for Synthetic Biology | Ulrik Gether Kenneth Lindegaard Madsen |

The funders had no role in study design, data collection and interpretation, or the decision to submit the work for publication.

### Author contributions

Simon Erlendsson, Conceptualization, Data curation, Formal analysis, Investigation, Visualization, Methodology, Writing—original draft, Writing—review and editing; Thor Seneca Thorsen, Conceptualization, Data curation, Formal analysis, Investigation, Methodology, Writing—review and editing; Georges Vauquelin, Conceptualization, Formal analysis, Validation, Investigation, Writing—original draft, Writing—review and editing; Ina Ammendrup-Johnsen, Volker Wirth, Data curation, Formal analysis, Visualization, Writing—review and editing; Karen L Martinez, Conceptualization, Supervision, Funding acquisition, Methodology, Writing—review and editing; Kaare Teilum, Supervision, Funding acquisition, Writing—original draft, Writing—review and editing; Ulrik Gether, Conceptualization, Supervision, Funding acquisition, Project administration, Writing—review and editing; Kenneth Lindegaard Madsen, Conceptualization, Supervision, Funding acquisition, Methodology, Writing—original draft, Project administration, Writing—review and editing

### Author ORCIDs

Simon Erlendsson http://orcid.org/0000-0002-6378-870X
Kaare Teilum http://orcid.org/0000-0001-6919-1982
Kenneth Lindegaard Madsen http://orcid.org/0000-0001-9274-6691

### Decision letter and Author response

Decision letter https://doi.org/10.7554/eLife.39180.024
Author response https://doi.org/10.7554/eLife.39180.025

## Additional files

### Supplementary files

• Transparent reporting form
DOI: https://doi.org/10.7554/eLife.39180.022

### Data availability

All data generated or analysed during this study are included in the manuscript and supporting files.

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
