## [Decision Letter]

Thank you for submitting your article "Mechanisms of PDZ domain scaffold assembly illuminated by use of supported cell membrane sheets" for consideration by eLife. Your article has been reviewed by three peer reviewers, and the evaluation has been overseen by a Reviewing Editor and Richard Aldrich as the Senior Editor. The following individual involved in the review of your submission has agreed to reveal their identity: Min Wu (Reviewer #2).

The reviewers have discussed the reviews with one another and the Reviewing Editor has drafted this decision to help you prepare a revised submission.

Summary:

This manuscript studies the interactions between the PDZ protein PICK1 and two previously-reported binding partners: the glutamate receptor GluA2 and the dopamine transporter DAT. Additional studies are shown characterizing the previously-reported interaction of another PDZ protein, PSD-95, with the β1- adrenergic receptor (β1AR). Binding affinities for these interactions are measured in two different ways: an in-solution fluorescence polarization assay and an assay measuring the fluorescence intensity of binding in supported cell membrane sheets (SCMS).

The authors find that binding strengths for PICK1, as well as other scaffolding protein (PDZ tandem domain from PSD-95), are orders of magnitude higher on SCMS than the affinities measured in solution binding assays (without membrane). The most interesting result is that the binding curves for PICK1 binding to a low affinity receptor and a high affinity receptor are two-fold different in total maximal binding, without changing overall Kd. They next develop a bivalent binding model that recapitulates well the experimental observations. By designing relevant mutants, they eventually perturbed the dimerization and successfully tested their model.

The manuscript has been carefully evaluated by three reviewers. They find that the experiments were rigorously performed and the data nicely quantified. Moreover, the results of the theoretical model are in qualitative agreement with the experimental results. They think that the findings are quite exciting and significant, since membrane protein interaction, especially interaction between multivalent proteins and membrane remains a poorly defined subject where solution-based assays for binding affinity still dominate. Although the theory for equilibrium binding of a bivalent ligand to cell surface receptor has been developed in the 80s (by Byron Goldstein), quantitative experimental work is limited. Moreover, this issue is clearly not widely appreciated, because one tends to assume that Kd is the half-maximal point without realizing that it only applies to simple binding reactions. Nevertheless, the reviewers agree that the manuscript could be further improved when different points that are listed below will be discussed in more details.

Essential revisions:

1) With the cell preparation (SCMS), many factors (other interacting proteins, lipids, etc.) can modify the protein-protein interaction to stabilize or otherwise modify it. The authors should discuss more extensively the origin of these differences (bulk versus SCMS) as well the possibility that with the SCMS preparation, different factors could perturb protein binding.

2) Different points in the theoretical model could be discussed:

- The theoretical model involves a system of differential equations to describe the binding kinetics, where phenomenological constants are used (Figure 4—figure supplement 1). Please discuss the values of these constants with respect to the binding energy and different type of forces involved in interaction of the ligand with the membrane (electrostatic forces, van der Waals forces, electric double layer forces, osmotic forces, hydration forces, mediated interaction forces etc. - see for example the book J.N. Israelachvili, Intermolecular and Surface Forces, Academic Press, London and Electrochimica Acta, 126: 42-60, 2014.)

- While describing their theoretical model, the authors refer to the model of Kramer and Karpen (1988) that also takes into account the allosteric effect. Could you discuss in the Discussion section how the allosteric effect is or could be described in the mathematical model presented in the manuscript (Figure 4—figure supplement 1), for example within a two-state model or some other model based on the statistical mechanics approach (by writing the corresponding partition function).

---

## [Author Response]

Essential revisions:1) With the cell preparation (SCMS), many factors (other interacting proteins, lipids, etc.) can modify the protein-protein interaction to stabilize or otherwise modify it. The authors should discuss more extensively the origin of these differences (bulk versus SCMS) as well the possibility that with the SCMS preparation, different factors could perturb protein binding.

We fully agree that the protein binding to SCMSs is affected by membrane/lipid binding as well as proteins native to the selected membrane. This together with the avidity was the rationale for undertaking the study as motivated in the Introduction, second paragraph. Causality in the study, however, is deduced from changes observed in the binding as a consequence to changes in the protein overexpressed in the SCMS and the purified proteins binding to it, with the membrane shape, composition and native protein expression remaining constant. Regardless, it is true that interactions with proteins native to the membrane and native membrane proteins may add significantly to the difference observed in binding strength on SCMS compared to in solution. We now open the Discussion with addressing this important point:

‘For PICK1, we sought to address the relative role of native constituents of the SCMSs to this increased binding strength. […] Conversely, we would argue that the PDZ interaction with transmembrane proteins native to the SCMS would be effectively competed by the overexpressed transmembrane proteins (given their comparable affinities e.g. Figure 1F and 3A) and consequently play a minor role in context of overexpression.’

2) Different points in the theoretical model could be discussed:- The theoretical model involves a system of differential equations to describe the binding kinetics, where phenomenological constants are used (Figure 4—figure supplement 1). Please discuss the values of these constants with respect to the binding energy and different type of forces involved in interaction of the ligand with the membrane (electrostatic forces, van der Waals forces, electric double layer forces, osmotic forces, hydration forces, mediated interaction forces etc. - see for example the book J.N. Israelachvili, Intermolecular and Surface Forces, Academic Press, London and Electrochimica Acta, 126: 42-60, 2014.)

The binding of PICK1 to SCMC’s is primarily determined by the interaction of the PDZ domain with the ligand and the nature of the interactions is well described from NMR/modeling. It primarily involve electrostatic interaction of the C-terminal carboxyl group of the ligand with the ‘carboxylate binding loop’ in the PICK1 PDZ domain (including K27 and D28) as well as hydrophobic interaction from the C-terminal residue (P0) and the third to last residue (P-2) of the ligand interacting with their respective hydrophobic pockets in the PDZ domain as described in ^1,2^. The affinity of this interaction as measured by in-solution fluorescence polarization binding was used to set the parameters k1 and k-1. The distance from r to determine the local concentration of the second PDZ domain in the monovalently bound state was determined from previous SAXS studies^3^. This was only stated in the figure legend previously and is now explicitly stated in the simulation section in Results (subsection “Simulations support dual binding modes for PICK1”). Thus *f* is the only truly phenomenological constant, which might include entropic penalty, steric hinderance etc. *f* is now described in more detail in the aforementioned subsection of the Results. Note that this value was increased to 185 after redoing the simulations (see below).

The binding by the amphipathic helix is driven by hydrophobic insertion in the membrane and electrostatic forces, and electrostatic binding of the BAR domain might also help binding although this was not experimentally addressed. These are not included in the model.

- While describing their theoretical model, the authors refer to the model of Kramer and Karpen (1988) that also takes into account the allosteric effect. Could you discuss in the Discussion section how the allosteric effect is or could be described in the mathematical model presented in the manuscript (Figure 4—figure supplement 1), for example within a two-state model or some other model based on the statistical mechanics approach (by writing the corresponding partition function).

Although ligand binding to the PICK1 PDZ domain has been described to relieve auto-inhibition of the BAR domain in PICK1^4^, our SAXS and membrane binding data does not support this notion^3^. Consequently, since there is limited evidence for an explicit allosteric component in the binding, this is included in the *f* factor in the present model. The Kramer and Karpen paper describes the binding of bi-valent PEG linked cGMP ligands to different receptors. Whereas the binding of cGMP results in an allosteric structural change in a neighboring subunit with a Hill coefficient of 2, the binding of the linked divalent ligands is non-cooperative with Hill coefficients around 1. Such an effect could explicitly be included our model by adding a parallel set of states where the PDZ domains have an altered affinity. However, as we have no experimental data to support such additional states this will be a purely theoretic discussion.

Alternatively, given the slow kinetics of the binding we initially also considered introducing a kinetic delay in the model, for mimicking how the PDZ binding could govern and facilitate subsequent insertion of the amphipathic helix. Similar to ‘kinetic proofreading’^5,6^ one might envision that a certain affinity (residence time might be needed to allow for the helix insertion. We tested binding of the PICK1 helix mutant to LKI and the maximal difference in maximal binding was preserved giving little experimental rationale for introduction of such a step.

These considerations have now been integrated in the discussion of the modeling (Discussion section).

1) Erlendsson, S. and Madsen, K.L. Membrane Binding and Modulation of the PDZ Domain of PICK1. Membranes (Basel) 5, 597-615 (2015).

2) Erlendsson, S. et al. Protein interacting with C-kinase 1 (PICK1) binding promiscuity relies on unconventional PSD-95/discs-large/ZO-1 homology (PDZ) binding modes for nonclass II PDZ ligands. J Biol Chem 289, 25327-40 (2014).

3) Karlsen, M.L. et al. Structure of Dimeric and Tetrameric Complexes of the BAR Domain Protein PICK1 Determined by Small-Angle X-Ray Scattering. Structure 23, 1258-70 (2015).

4) Lu, W. and Ziff, E.B. PICK1 interacts with ABP/GRIP to regulate AMPA receptor trafficking. Neuron 47, 407-421 (2005).

5) Hopfield, J.J. Kinetic proofreading: a new mechanism for reducing errors in biosynthetic processes requiring high specificity. Proc Natl Acad Sci U S A 71, 4135-9 (1974).

6) McKeithan, T.W. Kinetic proofreading in T-cell receptor signal transduction. Proc Natl Acad Sci U S A 92, 5042-6 (1995).